# Defining function of wild-type and three patient-specific *TP53* mutations in a zebrafish model of embryonal rhabdomyosarcoma

Jiangfei Chen[1,2†], Kunal Baxi[2,3†], Amanda E Lipsitt[2,4], Nicole Rae Hensch[2,3], Long Wang[2,3], Prethish Sreenivas[2,3], Paulomi Modi[2,3], Xiang Ru Zhao[2,3], Antoine Baudin[2,5], Daniel G Robledo[2], Abhik Bandyopadhyay[2], Aaron Sugalski[4], Anil K Challa[2,6], Dias Kurmashev[2], Andrea R Gilbert[7], Gail E Tomlinson[2,4], Peter Houghton[2,3], Yidong Chen[8], Madeline N Hayes[9], Eleanor Y Chen[10], David S Libich[2,5], Myron S Ignatius[2,3]*

[1]Institute of Environmental Safety and Human Health, Wenzhou Medical University, Wenzhou, China; [2]Greehey Children's Cancer Research Institute (GCCRI), UT Health Sciences Center, San Antonio, United States; [3]Department of Molecular Medicine, UT Health Sciences Center, San Antonio, United States; [4]Department of Pediatrics, Division of Hematology Oncology, UT Health Sciences Center, San Antonio, United States; [5]Department of Biochemistry and Structural Biology, UT Health Sciences Center, San Antonio, United States; [6]Department of Biology, University of Alabama at Birmingham, Birmingham, United States; [7]Department of Pathology and Laboratory Medicine, UT Health Sciences Center, San Antonio, United States; [8]Department of Population Health Sciences, UT Health Sciences Center, San Antonio, United States; [9]Developmental and Stem Cell Biology, Hospital for Sick Children, Toronto, Canada; [10]Department of Laboratory Medicine and Pathology, University of Washington, Seattle, United States

*For correspondence:
ignatius@uthscsa.edu

†These authors contributed equally to this work

Competing interest: The authors declare that no competing interests exist.

**Abstract** In embryonal rhabdomyosarcoma (ERMS) and generally in sarcomas, the role of wild-type and loss- or gain-of-function *TP53* mutations remains largely undefined. Eliminating mutant or restoring wild-type p53 is challenging; nevertheless, understanding p53 variant effects on tumorigenesis remains central to realizing better treatment outcomes. In ERMS, >70% of patients retain wild-type *TP53*, yet mutations when present are associated with worse prognosis. Employing a *kRAS*[G12D]-driven ERMS tumor model and tp53 null (tp53[-/-]) zebrafish, we define wild-type and patient-specific *TP53* mutant effects on tumorigenesis. We demonstrate that *tp53* is a major suppressor of tumorigenesis, where *tp53* loss expands tumor initiation from <35% to >97% of animals. Characterizing three patient-specific alleles reveals that *TP53*[C176F] partially retains wild-type p53 apoptotic activity that can be exploited, whereas *TP53*[P153Δ] and *TP53*[Y220C] encode two structurally related proteins with gain-of-function effects that predispose to head musculature ERMS. *TP53*[P153Δ] unexpectedly also predisposes to hedgehog-expressing medulloblastomas in the *kRAS*[G12D]-driven ERMS-model.

## Editor's evaluation

This paper uses the zebrafish as an in vivo model for exploring cancer genetics. The work on patient-specific alleles of the key oncogene TP53 enables new insights. The focus on embryonic stages enables a new understanding of the mechanism underlying this pediatric cancer.

## Introduction

*TP53* is the best-known tumor suppressor protein that is mutated or functionally disrupted in more than 50% of human tumors (*Kastenhuber and Lowe, 2017*; *Muller and Vousden, 2014*). Germline mutations in *TP53* are responsible for Li–Fraumeni syndrome that predisposes to a wide but distinct spectrum of tumors that varies with age (*Malkin, 2011*). Comprehensive analyses of *TP53* function in vivo and in vitro have revealed three different ways by which *TP53* can modulate tumorigenesis. These include effects caused by loss- or gain-of-function or dominant-negative mutations that disrupt the function of wild-type protein (*Ko and Prives, 1996*; *Levine et al., 1991*). Furthermore, the *TP53* mutation spectrum is likely tumor-specific (e.g., G>T transversions in lung cancer that do not correspond to the classic *TP53* hotspot mutations) (*Olive et al., 2004*; *Petitjean et al., 2007*). Recent studies looking at *TP53* alterations in rhabdomyosarcoma (RMS) patients show that *TP53* mutations are correlated with an increased risk of developing RMS and a worse prognosis (*Casey et al., 2021*; *Shern et al., 2021*); however, the role for specific *TP53* variants in sarcoma progression remains to be defined.

*TP53* mutations are present in >40% of adult carcinomas and thought to play a major role in tumorigenesis (*Gröbner et al., 2018*). In contrast, mutations in *TP53* are found in less than 6% of pediatric cancers (*Chen et al., 2014a*; *Chen, 2013*; *Gröbner et al., 2018*; *Seki et al., 2015*; *Shern et al., 2014*; *Tirode et al., 2014*). One major exception is Li–Fraumeni syndrome, where germline *TP53* mutations predispose individuals to a unique tumor spectrum that includes soft tissue sarcomas, such as RMS, and bone tumors (*Guha and Malkin, 2017*; *Malkin, 2011*). It is important to also note that the *TP53* pathway is often suppressed in sarcomas. For example, in human embryonal rhabdomyosarcoma (ERMS), a common pediatric cancer of muscle, the *TP53* locus is mutated or deleted in 16% of tumors, while transcriptional activity is altered in >30% of tumors, either through direct locus disruption or MDM2 amplification (*Chen, 2013*; *Seki et al., 2015*; *Shern et al., 2014*; *Taylor et al., 2000*). *TP53* mutations are also detected in ERMS at relapse, suggesting a role in tumor progression and/or resistance to therapy (*Chen, 2013*).

Mouse models have led the way in understanding *Tp53* function in vivo. Several murine genetic models were developed to assess the effects of both loss- and gain-of-function *Trp53* mutations (*Attardi and Donehower, 2005*; *Garcia and Attardi, 2014*) with mutant and null alleles spontaneously developing cancer in multiple tissues (*Lozano, 2010*). Of note, the tumor spectrum in mice varies depending on the mutant allele and genetic background; however, most in vivo studies have focused on a subset of hotspot mutations compared to a null or heterozygous background, and are seen only in a subset of sarcoma patients (*Attardi and Donehower, 2005*; *Garcia and Attardi, 2014*; *Guha and Malkin, 2017*). The vast majority of mutations observed in patients have not been interrogated in animal models, but rather function is inferred from these commonly studied hotspot mutants. *TP53* variants likely play different roles depending on tumor type, mutation, or the genetic background, presenting a significant challenge when defining pathogenicity and potential impacts on therapeutic approaches.

To define p53 biology in vivo, we recently generated a complete loss-of-function *tp53* deletion allele in syngeneic CG1-strain zebrafish using TALEN endonucleases (*Ignatius et al., 2018*). *tp53*^(del/del) (*tp53*^(-/-)) zebrafish spontaneously developed a spectrum of tumors that includes malignant peripheral nerve-sheath tumors (MPNSTs), angiosarcomas, germ cell tumors, and an aggressive natural killer cell-like leukemia (*Ignatius et al., 2018*). The role for *tp53* in self-renewal and metastasis of *kRAS*^(G12D)-induced ERMS tumors was assessed using cell transplantation assays and revealed that *tp53* loss does not change the overall frequency of ERMS self-renewing cancer stem cells compared to tumors expressing wild-type *tp53* (*Ignatius et al., 2018*). In contrast, *tp53*^(-/-) ERMS were more invasive and metastatic compared to *tp53* wild-type tumors, providing new insights into how *tp53* suppresses ERMS progression in vivo (*Ignatius et al., 2018*). In the present study, we take advantage of the fact that in zebrafish, similar to humans, p53 loss of function is not required for tumor initiation. We employ

$tp53^{-/-}$ zebrafish to further assess the role for p53 on $kRAS^{G12D}$-driven ERMS and find that wild-type $tp53$ is a major tumor suppressor in ERMS, affecting proliferation with a smaller effect on apoptosis. Next, we find that wild-type human $TP53$ is functional in zebrafish and potently suppresses ERMS initiation in vivo. We also define the pathogenicity of three $TP53$ mutations: (1) a $TP53^{C176F}$ variant found in ERMS patients (*Chen, 2013*; *Seki et al., 2015*; *Shern et al., 2014*); (2) a $TP53^{P153\Delta}$ variant of unknown significance, found in a teenager with an aggressive osteosarcoma at our clinic; and (3) a p53$^{Y220C}$ mutant that is structurally related to p53$^{P153\Delta}$ and may share some aspects of its function. We find that these three $TP53$ mutants have different effects on tumor initiation, location, proliferation, and apoptosis in our zebrafish model. Taken together, these analyses highlighting a role for the zebrafish ERMS model that can be used to characterize the spectrum of both common and rare $TP53$ mutations in sarcoma patients.

## Results

### *tp53* is a potent suppressor of ERMS initiation, growth, and invasion

Whole-genome-/-exome-/-transcriptome sequencing analysis revealed that a majority of primary ERMS tumors are wild type for TP53. However, TP53 loss, mutation, or MDM2 amplification accounts for pathway disruption in approximately 30% of ERMS (*Chen, 2013*; *Seki et al., 2015*; *Shern et al., 2014*). Furthermore, TP53 mutations are associated with poor prognosis (*Casey et al., 2021*; *Shern et al., 2021*). In the zebrafish ERMS model, expressing the human kRAS$^{G12D}$ oncogene in muscle progenitor cells is sufficient to generate ERMS with morphological and molecular characteristics that recapitulate the human disease (*Langenau et al., 2007*). Although robust, in a tp53 wild-type genetic background, the upper limit of tumor formation observed is 40% by 50 d post injection/fertilization (*Langenau et al., 2007*). To test whether tp53 suppresses ERMS initiation in vivo, we generated tumors using rag2:KRAS$^{G12D}$ in tp53 wild-type (tp53$^{+/+}$) and tp53$^{-/-}$ mutant zebrafish (*Figure 1—figure supplement 1*). Co-injection of rag2:kRAS$^{G12D}$ and rag2:DsRed resulted in approximately 34% of tp53$^{+/+}$ animals (n = 49/143 animals) with DsRed-positive tumors by 60 d post injection. In stark contrast, up to 97% of tp53$^{-/-}$ animals displayed tumors (n = 139/142 animals) by 60 d. kRAS$^{G12D}$-induced tp53$^{-/-}$ ERMS displayed a very rapid onset, with the majority of arising within the first 20 d of life (*Figure 1A*; p<0.0001). We assessed tumor size and the number of tumors initiated and found that tp53$^{-/-}$ mutant ERMS displayed a significant increase in size (*Figure 1B and D*; p<0.0001) and in the number of tumors per animal (*Figure 1B and C*; p<0.0001) compared to tp53$^{+/+}$ animals. Histological assessment showed no major differences between tp53$^{+/+}$ and tp53$^{-/-}$ ERMS, and tumors formed in the head, trunk, or tail similarly in both backgrounds (*Figure 1E and F*, *Figure 1—figure supplement 2B*). We assessed relative expression of kRAS$^{G12D}$ and its downstream effector dusp4 in tumors from tp53$^{-/-}$ and tp53$^{+/+}$ zebrafish and confirmed similar expression levels of kRAS$^{G12D}$ and dusp4 using quantitative qPCR (*Figure 1—figure supplement 2A*; n = 3 tumors each, p=0.8966 unpaired *t*-test with Welch's correction). Taken together, our analysis suggests that *tp53* is a major suppressor of tumor initiation in RAS-driven zebrafish ERMS.

### *tp53* suppresses proliferation and to a lesser extent apoptosis in ERMS tumors

Given that *tp53*$^{-/-}$ tumors are larger than equivalent staged *tp53*$^{+/+}$ tumors, we performed EdU and phospho-histone H3 staining to assess proliferation and Annexin V staining to assess apoptosis in primary ERMS. Wild-type and *tp53*$^{-/-}$ tumor-burdened animals were treated with a 6 hr pulse of EdU, euthanized, sectioned, and stained for EdU-positive cells, a marker for proliferation. EdU analyses revealed that *tp53*$^{-/-}$ tumors displayed a significant increase in EdU-positive cells (*Figure 2A*; p<0.0001 Student's *t*-test) and significantly more phospho-histone H3-expressing cells when compared to *tp53* wild-type ERMS sections (*Figure 2B*; p<0.01 Student's *t*-test), indicating increased proliferation. Annexin V staining, that can distinguish cells beginning to undergo apoptosis (early apoptosis; low Annexin V/ propidium iodide [PI]-positive), or that are undergoing apoptosis (late apoptosis; high Annexin V/ high PI) or necrosis (high Annexin V/ low PI), was performed on live single-cell suspensions of ERMS tumor cells extracted post euthanasia. Annexin V staining showed no significant difference in the rate of late apoptosis (Q2) (*Figure 2C–E*; p=0.592 Student's *t*-test), but showed a small yet significant decrease in early apoptosis (Q4) in *tp53*$^{-/-}$ mutant ERMS compared to wild-type (*Figure 2C–E*;

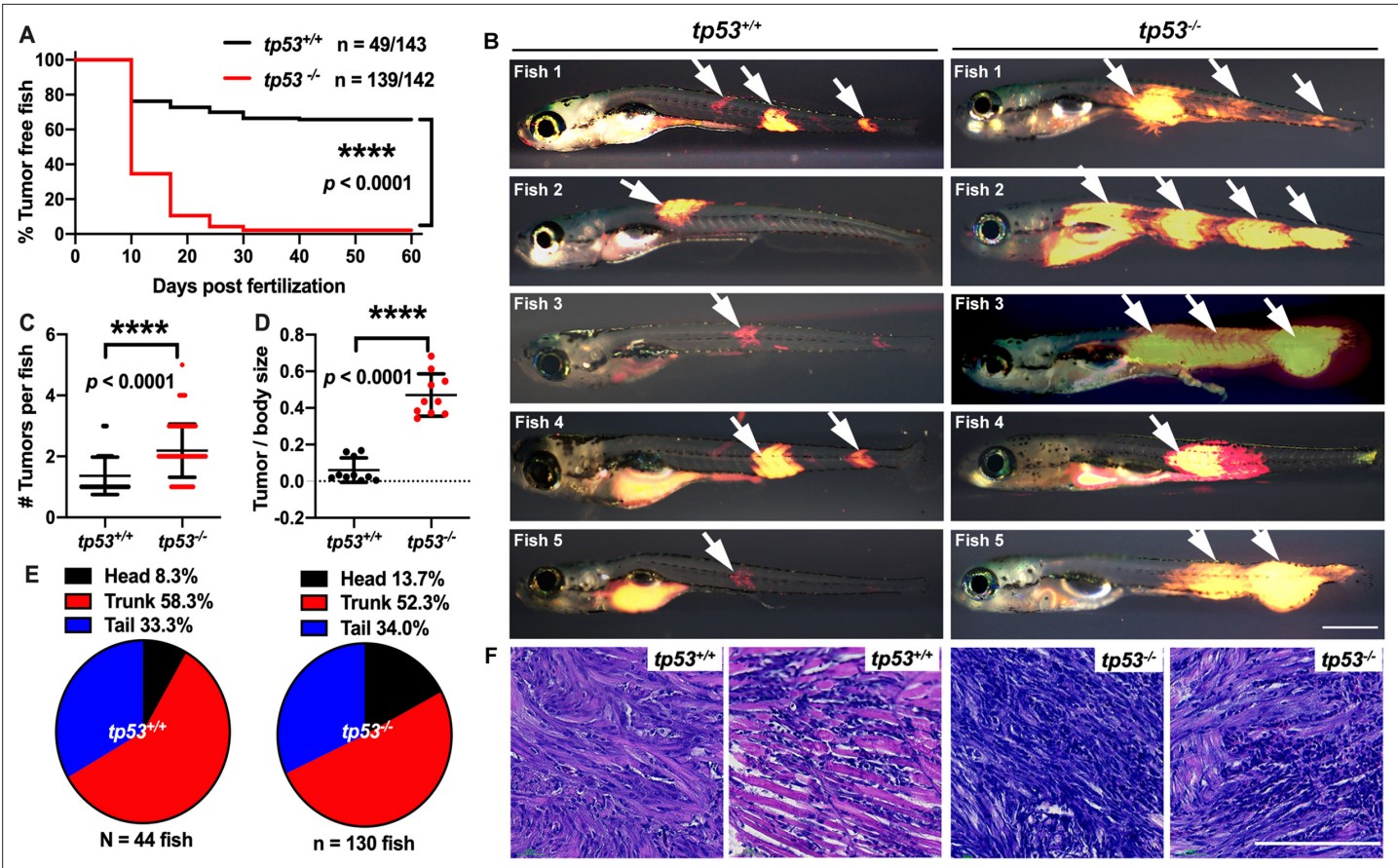

**Figure 1.** *tp53* suppresses embryonal rhabdomyosarcoma (ERMS) tumor initiation. (**A**) Kaplan–Meier plot showing ERMS tumor initiation in *tp53*-/- and *tp53*+/+ fish. (**B**) Representative images of DsRed-positive zebrafish ERMS. Arrows show tumor location for each fish. All tumor-burdened zebrafish are 10 days old. Scale bar = 0.5 mm. (**C**) Tumor numbers per zebrafish in *tp53*-/- and *tp53*+/+ fish. n = 44 (*tp53*+/+), n=130 (*tp53*-/-). (**D**) Ratio of tumor area to total body area in in *tp53*-/- and *tp53*+/+ fish. n = 10. (**E**) Pie chart showing percentage of tumors found in varying regions of *tp53*-/- and *tp53*+/+ fish, showing no significant differences in tumor localization. Head – p=0.25848, trunk – p=0.39532, tail – p=0.92034 (two-tailed two proportions *Z*-test). (**F**) Representative H&E staining of zebrafish ERMS tumors. Scale bar = 100 µm.

The online version of this article includes the following figure supplement(s) for figure 1:

**Figure supplement 1.** Schematic of the experimental setup to generate embryonal rhabdomyosarcoma (ERMS) tumors in zebrafish via microinjection of indicated linearized DNA constructs into the one-cell-stage zebrafish embryos generated from in crosses of *tp53*-/- or *tp53*+/+ zebrafish.

**Figure supplement 2.** tp53 suppresses embryonal rhabdomyosarcoma (ERMS) tumor initiation.

p=0.0073 Student's *t*-test). Taken together, our data suggests that *tp53* is a potent suppressor of kRAS-induced ERMS tumor cell proliferation, with only a moderate effect on apoptosis, affecting ERMS initiation.

## Reintroducing human *TP53* in *tp53*-/- zebrafish blocks tumor initiation, growth, proliferation, and increases apoptosis

Zebrafish and human p53 are functionally similar and share 56% identity with respect to amino acid sequence (67% positives, *Figure 3—figure supplement 1*; *Berghmans et al., 2005*; *Ignatius et al., 2018*; *Parant et al., 2010*; *Storer and Zon, 2010*). Within the core DNA-binding region where a majority of mutations occur in patients, 72% conservation exists (79% positives, *Figure 3—figure supplements 1 and 2*). To assess functional conservation in vivo, we co-expressed wild-type human *TP53* (*TP53*WT) in the cells from which ERMS tumors initiate in *tp53*-/- animals (*Figure 3—figure supplement 3*). Importantly, co-expression of *kRAS*G12D along with *TP53*WT resulted in the expression of wild-type *TP53* from the very beginning in cells from which tumors initiate and also in the resulting tumors. Co-expression of *kRAS*G12D along with *TP53*WT significantly suppressed tumor initiation (*Figure 3A*;

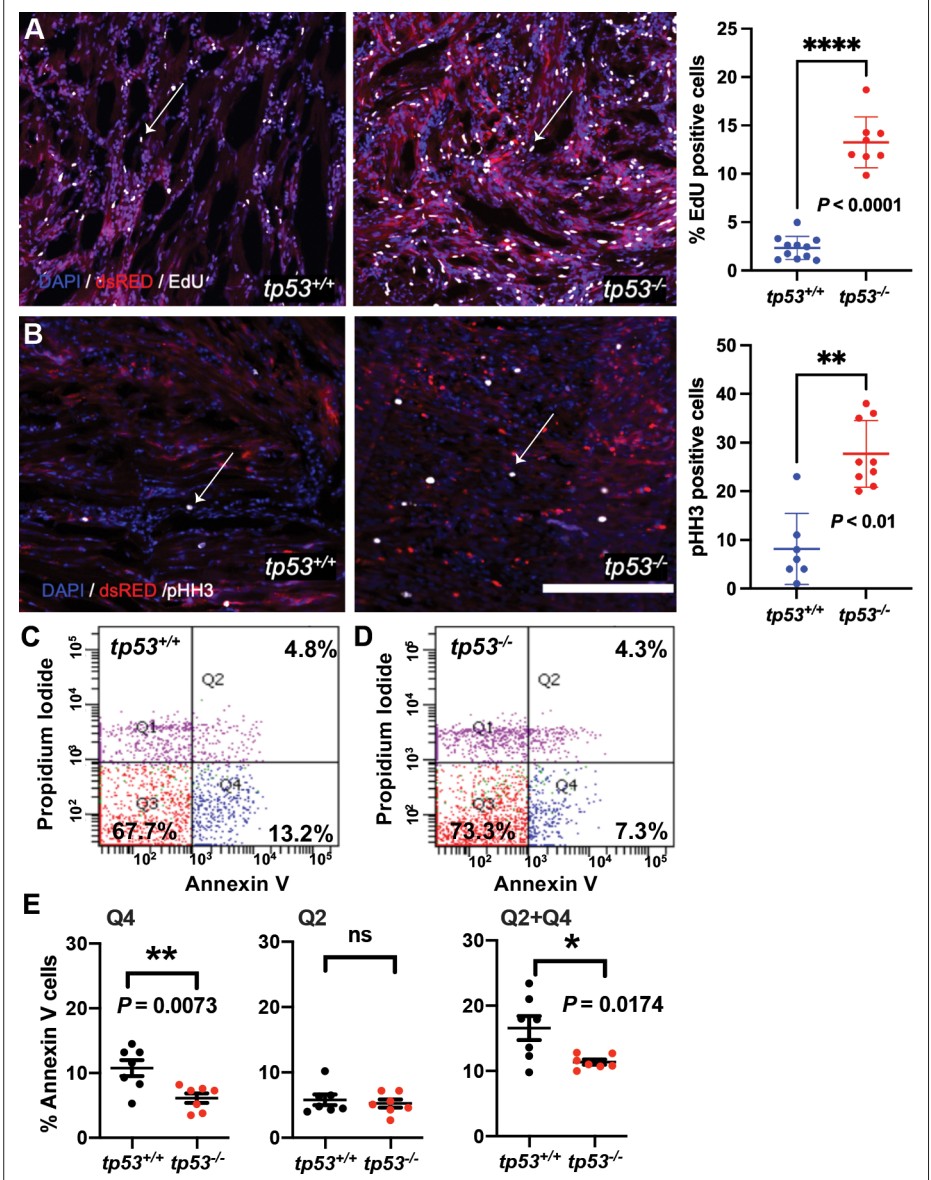

**Figure 2.** *tp53* is a potent suppressor of proliferation and to a lesser extent of apoptosis. (**A**) Representative confocal microscopy images of EdU staining on embryonal rhabdomyosarcoma (ERMS) tumor sections and a plot quantifying the percentage of EdU-positive cells. Average of n = 8–11/primary tumors. White arrows show EdU-positive cells. (**B**) Representative confocal microscopy images of phospho-histone H3 staining on ERMS tumor sections (scale bar = 100 μm). Total number pHH3-positive cells per single ERMS tumor ×200 confocal image section assessed from n7-9 primary tumors. One the right-most panel is a plot quantifying the total number of pHH3-positive cells per single ERMS section. White arrows show pHH3-positive cells. (**C, D**) Representative flow cytometry analysis of Annexin V staining of *tp53⁺/⁺* and *tp53⁻/⁻* ERMS tumors, respectively. (**E**) Quantification of flow cytometry analysis of Annexin V staining. Q1 = pre-necrotic cells, Q2 = late apoptosis + necrotic cells, Q3 = living cells, Q4 = early apoptotic cells. n = 7. ns, not significant, p=0.5926, unpaired *t*-test.

p<0.0001, Student's *t*-test). ERMS in *TP53^WT*-expressing animals were smaller than tumors in age-matched *tp53⁻/⁻* zebrafish (***Figure 3B and C***; p<0.0001 Student's *t*-test); however, we observed no significant difference in the number or distribution of ERMS initiated per zebrafish between the two groups (***Figure 3D and E***; p=0.065 for ***Figure 3D***, Student's *t*-test). Analysis of EdU staining of ERMS tumors determined that co-expressing *TP53^WT* significantly inhibited proliferation (***Figure 3F***; p=0.0007 Student's *t*-test). Also, we observed an overall increase in the number of apoptotic cells in *TP53^WT*-expressing tumors, as seen by Annexin V staining (***Figure 3G***; p<0.0001 Student's *t*-test).

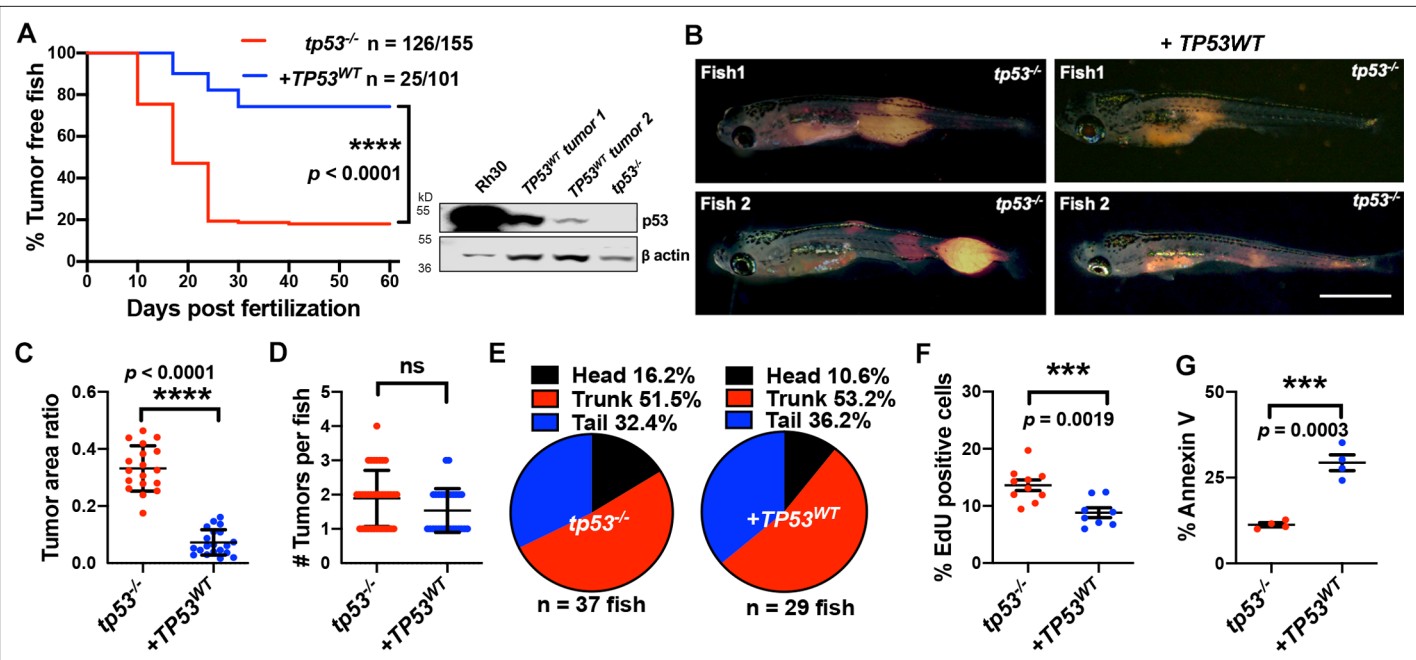

**Figure 3.** Human *TP53* blocks tumor initiation, growth, and proliferation and increases apoptosis in *tp53⁻/⁻* zebrafish. (**A**) Kaplan–Meier plot showing embryonal rhabdomyosarcoma (ERMS) tumor initiation in *tp53⁻/⁻* fish with or without p53^WT expression. Western blot analysis was performed to assess p53^WT expression level in tumors. (**B**) Representative images of ERMS tumors in *tp53⁻/⁻* fish with or without human *TP53^WT* expression. Tumor-burdened zebrafish are between 15 and 20 days old. Scale bar = 1 mm. (**C**) Ratio of tumor area to total body area in *tp53⁻/⁻* fish with or without expression of *TP53^WT*. n = 18. (**D**) Number of tumors per *tp53⁻/⁻* zebrafish with or without expression of *TP53^WT*. ns, not significant. n = 36 (*tp53⁻/⁻*), n = 28 (*TP53^WT*). (**E**) Pie chart showing site of tumor localization in *tp53⁻/⁻* fish with or without expression of *TP53^WT* showing no statistical differences. Head – p=0.20045, trunk – p=0.42858, tail – p=0.3336. Quantification of proliferation (**F**) and apoptosis (**G**) via EdU staining (n = 10) and Annexin V staining (n = 3), respectively, for tumors arising in *tp53⁻/⁻* fish with or without expression of *TP53^WT*.

The online version of this article includes the following figure supplement(s) for figure 3:

**Figure supplement 1.** Human TP53 blocks tumor initiation, growth, and proliferation and increases apoptosis in tp53⁻/⁻ zebrafish.

**Figure supplement 2.** Human TP53 blocks tumor initiation, growth, and proliferation and increases apoptosis in tp53⁻/⁻ zebrafish.

**Figure supplement 3.** Schematic of the experimental setup to generate embryonal rhabdomyosarcoma (ERMS) tumors in zebrafish via microinjection of indicated linearized DNA constructs into the one-cell-stage zebrafish embryos generated from in crosses of *tp53⁻/⁻* zebrafish.

**Figure supplement 4.** Zebrafish tp53 (italics) blocks tumor initiation in tp53⁻/⁻ zebrafish.

Similarly, wild-type zebrafish *tp53* when co-expressed with *kRAS^G12D* in *tp53⁻/⁻* zebrafish significantly suppressed tumor initiation, proliferation, and apoptosis (**Figure 3—figure supplement 4**; p<0.0001 Student's *t*-test). Importantly, expression of human *TP53^WT* or wild-type zebrafish *tp53* selectively in ERMS using the *rag2* promoter had no effect on the overall viability of zebrafish embryos or larvae. Co-expression of human *TP53* or zebrafish *tp53* in *tp53⁻/⁻* zebrafish did result in increased apoptosis compared to tumors generated in *tp53⁺/⁺* zebrafish. However, it is important to point out that in this assay human p53 was expressed at significantly lower levels than is present in an RMS cell line expressing mutant p53 (see **Figure 3A**).

## *TP53^C176F* is a hypomorphic allele while *TP53^P153Δ* has gain-of-function effects in ERMS

The effects of specific *TP53* mutations on ERMS are not well defined. Analysis of *TP53* mutations in RMS patients found that a majority of mutations lie outside the well-studied hotspot locations and are mostly uncharacterized (**Figure 3—figure supplement 2**). To determine whether ERMS-expressing patient-specific *TP53* point mutations differ from *TP53* deletion in vivo, we assessed mutant activity in *tp53⁻/⁻* (null) ERMS. The first selected mutation was a *TP53^C176F* allele that is expressed by at least two ERMS patients (**Chen, 2013**; **Seki et al., 2015**). In one of these patients, the second *TP53* allele was deleted in the tumor and *TP53^C176F* is present in both the primary and relapsed tumor (**Chen,**

*2013*). Another mutation we selected to model was a rare *TP53^P153Δ* (deletion of Proline 153, P153Δ) allele present in a patient in our clinic with an aggressive osteosarcoma. This patient developed an aggressive refractory osteosarcoma as a teenager, while her mother developed osteosarcoma in her early twenties; both eventually succumbed to their disease. Based on the patient's family history, tumor type, and *TP53* mutation, a diagnosis of Li–Fraumeni syndrome was made. However, since the mutation is rare, pathogenicity could not be assigned in this particular allele.

Analysis of amino acid sequence conservation showed that p53 residue C176 is conserved across humans, mice, and zebrafish. P153 is present only in humans; however, it is located in a region that is highly conserved across all three species and P153 is conserved across other closely related mammalian species (*Figure 4A*, *Figure 3—figure supplement 2*). We confirmed *TP53^P153Δ* mutation by sequencing the DNA from the first passage murine patient-derived xenograft (PDX) generated from the patient at autopsy. Tumor cells obtained from the patient also harbored a missense mutation at c.476C>T (p.A159V) (*Figure 4B*). We next assessed whether p53^C176F and p53^P153Δ proteins were expressed in the ERMS murine PDX SJRHB00011 that harbors the *TP53^C176F* mutation and in the primary osteosarcoma that harbors the *TP53^P153Δ* mutation using immunohistochemistry on tumor sections (*Chen, 2013*). Both the SJRHB00011 PDX tumor and the primary osteosarcoma tumor expressed p53 protein, as evidenced by strong positive nuclear staining in the majority of tumor cells (*Figure 4C and D*). H&E staining confirmed RMS (heterogeneous population of ovoid to slightly spindled cells; *Figure 4E*) and osteosarcoma diagnosis (pleomorphic neoplastic tumor cells with irregular disorganized trabeculae of unmineralized malignant osteoid; *Figure 4F*).

We generated ERMS in *tp53^-/-* zebrafish using the same approach as earlier described and co-expressed *TP53^C176F* or *TP53^P153Δ* along with *kRAS^G12D* in *tp53^-/-* embryos in cells from which tumors initiate (*Figure 4—figure supplement 1*). Expression of mutant p53 protein in ERMS tumors was confirmed by western blot analysis using a human-specific p53 antibody. Mutant p53 expression in Rh30 RMS cells was used as a positive control while ERMS cells from *tp53^-/-* zebrafish were used as a negative control (*Figure 4G*). Expression of *TP53^C176F* with *kRAS^G12D* in *tp53^-/-* zebrafish resulted in a significant reduction in tumor initiation compared to expressing *kRAS^G12D* alone (*Figure 4H*, p=0.0005). By assessing *TP53^C176F*-expressing zebrafish for ERMS incidence, location, proliferation, and apoptosis, we found that the number of primary ERMS per fish was reduced but not significant between *tp53^-/-* and *tp53^-/-* + *TP53^C176F*-expressing animals (*Figure 4—figure supplement 2A*, *tp53^-/-* vs. *tp53^-/-* + *TP53^C176F*, p=0.065 (*TP53^C176F*) one-way ANOVA with Tukey's multiple-comparisons test). *tp53^-/-* + *TP53^C176F*-expressing ERMS had fewer EdU-positive cells but similar levels of cells undergoing mitosis compared to *tp53^-/-* tumors (*Figure 4—figure supplement 2B and C*). In contrast, *TP53^C176F* tumors showed significantly higher rates of apoptosis (*Figure 4K*, p<0.0001). There were no histological differences between the two groups (*Figure 4L*).

In contrast to *TP53^C176F*-expressing ERMS, expression of *TP53^P153Δ* with *kRAS^G12D* in *tp53^-/-* zebrafish did not affect tumor initiation compared to *kRAS^G12D* alone (*Figure 4H*, p=0.774). However, unexpectedly, *tp53^-/-* + *TP53^P153Δ*-expressing zebrafish had developed twice as many ERMS in the head musculature (*Figure 4I and J*, p=0.0096, two proportions *Z*-test), suggesting a gain-of-function effect with respect to initiation site. Additionally, while we did not observe a difference in apoptosis, we did find that *tp53^-/-* + *TP53^P153Δ* tumors were less proliferative compared to *tp53^-/-* (*Figure 4—figure supplement 2B and C*). Tumor histology remained unchanged across both groups (*Figure 4L*).

To assess the effects of p53 variant expression on transcription, we performed qPCR analyses comparing *tp53^+/+*, *tp53^-/-* and *tp53^-/-* + *TP53^C176F* and *tp53^-/-* + *TP53^P153Δ* ERMS and p53 target genes including *baxa*, *cdkn1a*, *gadd45a*, *noxa*, and *puma/bbc3*. We found that both *tp53^-/-* + *TP53^C176F* and *tp53^-/-* + *TP53^P153Δ* tumors displayed increased *bbc3* expression compared to *tp53^-/-* or *tp53^+/+* tumors, but had no difference with respect to *cdkn1a*, *gadd45a* and *noxa* expression. We also found that *tp53^-/-* + *TP53^C176F* and *tp53^-/-* + *TP53^P153Δ* tumors expressed higher levels of *cdkn1a* compared to *tp53^+/+* ERMS; however, rather unexpectedly *tp53^-/-* tumors also express higher levels of *cdkn1a* compared to *tp53^+/+* controls (*Figure 4—figure supplement 3*).

Given that the *TP53^C176F* mutation retains the ability to induce apoptosis, we next tested whether this activity could be augmented to inhibit tumor growth in vivo. It has been shown previously that ZMC1, a synthetic metallochaperone that transports zinc into cells as an ionophore, can restore p53 activity by stabilizing mutant p53 proteins such as p53^C176F (*Blanden et al., 2015*). To test this, we generated ERMS in the syngeneic CG1 strain zebrafish that were either *tp53^-/-* or *tp53^-/-* +*TP53^C176F*.

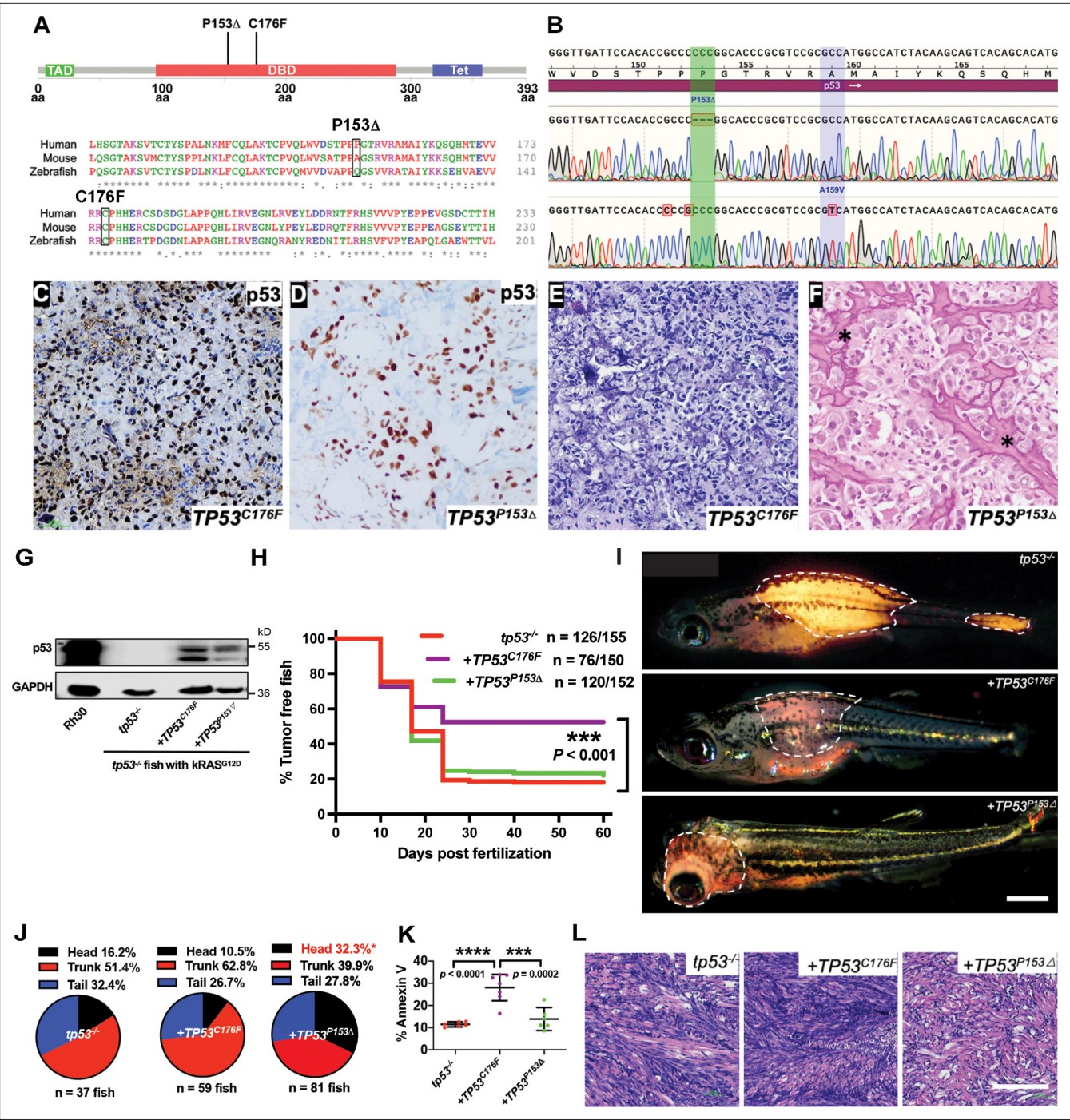

**Figure 4.** Assigning pathogenicity to two human *TP53* sarcoma mutations in the kRAS^G12D^-driven embryonal rhabdomyosarcoma (ERMS) model.
(**A**) Lollipop plot showing the two novel, human p53 mutations P153Δ and C176F, as well as the amino acid sequence alignment for human, mouse, and zebrafish protein. (**B**) DNA sequencing data from osteosarcoma patient confirming the germline P153Δ mutation, as well as somatic A159V mutation. (**C**, **D**) p53 immunohistochemistry staining of p53 in ERMS patient-derived xenograft (PDX) SJRHB00011 expressing p53^C176F^ and osteosarcoma expressing p53^P153Δ^. (**E**, **F**) Representative H&E staining of ERMS PDX expressing the C176F mutation and diagnostic biopsy of osteosarcoma tumor expressing osteosarcoma expressing p53^P153Δ^ showing neoplastic tumor cells with pleomorphic nuclei, irregular chromatin pattern, as well as irregular disorganized trabeculae of unmineralized malignant osteoid (stars). (**G**) Protein expression of mutant p53 in zebrafish ERMS tumors, with rhabdomyosarcoma (RMS) cell line, Rh30, as a control. (**H**) Kaplan–Meier plot showing tumor initiation in *tp53*^-/-^ fish with or without expression of mutant *TP53*. (**I**) Representative images of tumor localization in *tp53*^-/-^ fish with or without expression of mutant *TP53*. Age of zebrafish in panels is 37 d. Scale bar = 1 mm (**J**) Pie chart showing percentage of tumors found in varying regions of *tp53*^-/-^ fish with or without expression of mutant *TP53*. Percentages in red indicate a significant difference to *tp53*^-/-^ (p=0.0096, two-tailed two proportions *Z*-test). (**K**) Quantification of Annexin V staining in tumors arising in *tp53*^-/-^ fish with or without

*Figure 4 continued on next page*

*Figure 4 continued*

expression of mutant *TP53*. n = 6–7. (**L**) Representative H&E staining of tumors arising in *tp53$^{-/-}$* fish with or without expression of mutant *TP53*. Scale bar = 100 µm.

The online version of this article includes the following figure supplement(s) for figure 4:

**Figure supplement 1.** Schematic of the experimental setup to generate embryonal rhabdomyosarcoma (ERMS) tumors in zebrafish via microinjection of indicated linearized DNA constructs into the one-cell-stage zebrafish embryos generated from in crosses of *tp53$^{-/-}$* zebrafish.

**Figure supplement 2.** Assigning pathogenicity to two human TP53 sarcoma mutations in the kRASG12D-driven embryonal rhabdomyosarcoma (ERMS) model.

**Figure supplement 3.** Semi-quantitative qPCR analyses comparing expression of known p53 direct regulated genes *baxa, bbc3, cdkn1a, gadd45a,* and *noxa* in primary zebrafish embryonal rhabdomyosarcoma (ERMS) expressing wt *tp53* (blue, n = 3), *tp53$^{-/-}$* (red, n = 3), *tp53$^{-/-}$ + TP53$^{C176F}$* (purple, n = 4), or *tp53$^{-/-}$ + TP53$^{P153Δ}$* (green, n = 4).

Next, we expanded tumors from both groups in recipient wild-type CG1 animals and treated tumors in recipient host animals for 2 wk with either DMSO or ZMC1. Compared to DMSO-only control treatment, *tp53$^{-/-}$ + TP53$^{C176F}$*-expressing ERMS treated with ZMC1 showed a significant reduction in tumor growth over time (***Figure 4—figure supplement 2D and G***; p=0.0116 at week 3). We next assessed effects of ZMC1 on apoptosis and found increased apoptosis in *tp53$^{-/-}$ + TP53$^{C176F}$*-expressing ERMS treated with ZMC1 (***Figure 4—figure supplement 2F***; p=0.0008). ZMC1 treatment increased p53$^{C176F}$ expression at the protein level, suggesting increased stability of p53$^{C176F}$ protein in response to drug (***Figure 4—figure supplement 2E***). In contrast, ZMC1 treatment in *tp53$^{-/-}$* ERMS cohorts did not affect tumor growth (***Figure 4—figure supplement 2H and K***, p=0.961) or apoptosis (***Figure 4—figure supplement 2I and J***, p=0.583).

Altogether, *TP53$^{C176F}$* appears to retain some wild-type function that partially prevents ERMS initiation in *tp53$^{-/-}$* zebrafish, as well as trigger apoptosis in vivo. However, this variant appears not to have a significant effect on cell proliferation. Interestingly, the activity of p53$^{C176F}$ can be further enhanced by tp53 reactivators, such as ZMC1. In contrast, we found that the *TP53$^{P153Δ}$* mutation functions as a gain-of function allele with respect to the site of tumor initiation, but has no effect on overall tumor initiation. However, tumors expressing the *TP53$^{P153Δ}$* while having no effects on apoptosis compared to *tp53$^{-/-}$* have significantly fewer proliferating cells, suggesting an allele with gain of function and also some wild-type activity (***Supplementary file 1***).

## Expression of *TP53$^{P153Δ}$* with *kRAS$^{G12D}$* in *tp53$^{-/-}$* zebrafish results in the initiation of medulloblastomas with a shh gene signature

The head ERMS tumors that are formed in zebrafish expressing *kRAS$^{G12D}$* and *TP53$^{P153Δ}$* in the tp53$^{-/-}$ background initiate tumors in the head musculature around the eye and in the jaw; however, these tumors are overall less proliferative compared to the trunk tumors (***Figure 4—figure supplement 2B and C***). We generated tumors expressing *kRAS$^{G12D}$* and *GFP* or *kRAS$^{G12D}$, TP53$^{P153Δ}$,* and *GFP* in syngeneic *tp53$^{-/-}$* CG1 strain zebrafish and expanded three primary head tumors from *kRAS$^{G12D}$; TP53$^{P153Δ}$;tp53$^{-/-}$; GFP*-expressing animals and one head and two trunk tumors expressing control *kRAS$^{G12D}$; tp53$^{-/-}$; GFP* in syngeneic CG1 strain recipients that are wild-type for *tp53*. We find no difference in labeling tumors with different fluorescent reporter proteins, so we use these reporters interchangeably (***Ignatius et al., 2012***; ***Ignatius et al., 2018***; ***Langenau et al., 2008***). Secondary tumor cells were enriched by flow cytometry and RNA extracted and processed for RNA sequencing analyses. RNAseq unexpectedly revealed that all three *kRAS$^{G12D}$;TP53$^{P153Δ}$; tp53$^{-/-}$;GFP-positive* tumors displayed RNA expression signatures associated with neuronal differentiation, neurotransmitter secretion, synapse formation, and neurogenesis; while *kRAS$^{G12D}$; tp53$^{-/-}$;GFP-positive*-only tumors as expected displayed signatures associated with myogenesis, skeletal muscle structure, and muscle differentiation (***Figure 5A and B***, ***Supplementary file 2***). Unbiased gene set enrichment analysis (GSEA) found that *TP53$^{P153Δ}$* tumors had gene signatures associated with pro-neural glioblastoma, neuronal markers, and the downregulation of negative regulators of hedgehog signaling. These tumors also displayed negative correlation with soft tissue tumors and glioblastomas of mesenchymal origin (***Figure 5—figure supplement 1***, ***Supplementary file 2***). Together, these data suggest that at least some of the head tumors are more likely brain tumors with neuronal gene expression signatures, rather than ERMS.

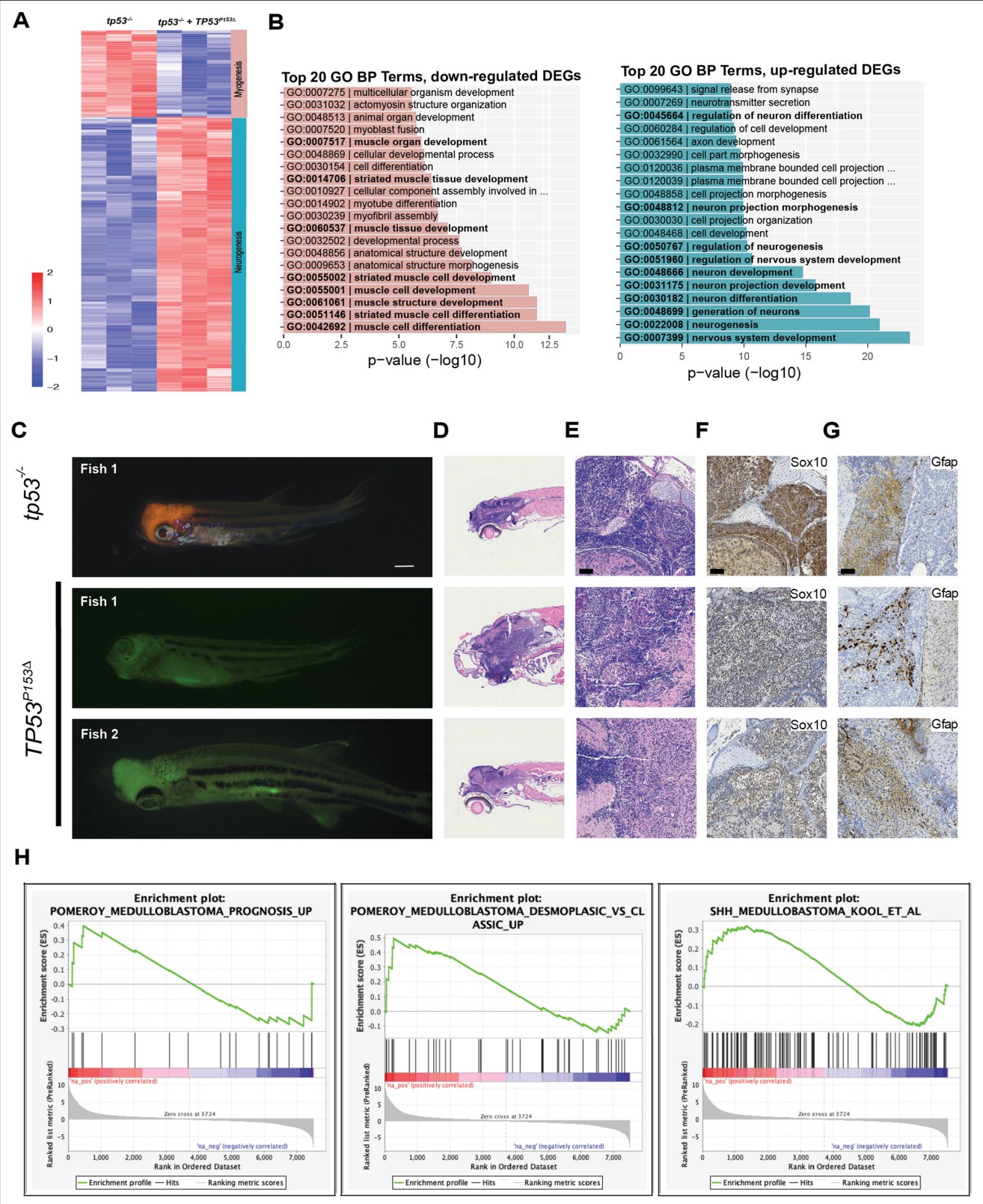

**Figure 5.** Expression of *TP53^P153Δ^* with *kRAS^G12D^* in *tp53^-/-^* zebrafish results in the initiation of medulloblastomas with a shh gene signature. (**A**) Heatmap from RNAseq analyses comparing tumors expressing *kRAS^G12D^*;*tp53^-/-^* to *kRAS^G12D^*; *tp53^-/-^*; *TP53^P153Δ^* (n = 3/group). A total of 643 genes were selected for the heatmap, with adjusted p-value<0.01 and fold-change >10. (**B**) Enriched Gene Ontology (GO) Biological Processes (BP) in upregulated genes in p53-/- group (left panel with pink bars) consistent with the expected tissue of origin for *kRAS^G12D^*;*tp53^-/-^* to *kRAS^G12D^*; *tp53^-/-^*; *TP53^P153Δ^* (right panel

*Figure 5 continued on next page*

*Figure 5 continued*

with light blue bars). (**C**) Representative images of medulloblastoma tumors expressing *kRAS*$^{G12D}$ and DsRED (top panel) or GFP (bottom two panels) in *tp53*$^{-/-}$ fish with or without human *TP53*$^{P153\Delta}$ expression. Tumor-burdened zebrafish are between 30 (Fish 1) and 70 (Fish 2, 3) days old. Scale bar = 1 mm. (**D**) Representative H&E staining of tumors arising in the head region of *tp53*$^{-/-}$ fish with or without expression of human TP53$^{P153\Delta}$. Images between 40 and 60 times magnification. (**E–G**) Representative sections of tumor-burdened zebrafish showing H&E staining (**D, E**) and IHC staining for Sox10 (**F**), and Gfap (**G**) in head tumors expressing *kRAS*$^{G12D}$ in *tp53*$^{-/-}$ fish or *kRAS*$^{G12D}$ and*TP53*$^{P153\Delta}$ in tp53$^{-/-}$ fish. Scale bar = 60 μm (**H**) Gene set enrichment analysis (GSEA) showing the enrichment of medulloblastoma gene signatures from Pomeroy et al., and Kool et al., with our zebrafish brain tumors. Log-fold-change derived from the differential expression analysis between *tp53*$^{-/-}$ and *tp53*$^{-/-}$; *TP53*$^{P153\Delta}$ was used with GSEA pre-ranked function (GSEA, v4.0.3, Broad Institute, MA).

The online version of this article includes the following figure supplement(s) for figure 5:

**Figure supplement 1.** Unbiased gene set enrichment analysis (GSEA) (GSEApreranked function) comparing gene expression between *tp53*$^{-/-}$ to *tp53*$^{-/-}$; *TP53*$^{P153\Delta}$ tumors (log-fold-change) were performed with chemical and genetic perturbation (CGP) gene sets, part of MSigDB C2 curated gene set collection.

**Figure supplement 2.** Schematic of the experimental setup to generate embryonal rhabdomyosarcoma (ERMS) tumors in zebrafish via microinjection of indicated linearized DNA constructs into the one-cell-stage zebrafish embryos generated from in crosses of *tp53*$^{-/-}$ zebrafish.

**Figure supplement 3.** Expression of *TP53*$^{P153\Delta}$ with kRAS$^{G12D}$ in tp53$^{-/-}$ zebrafish results in the initiation of medulloblastomas with a shh gene signature. (*TP53*$^{P153\Delta}$).

**Figure supplement 4.** Gene set enrichment analysis (GSEA) showing the enrichment of medulloblastoma gene signatures from Pomeroy et al., with our zebrafish brain tumors.

To further assess the unexpected finding of tumors of neural origin in the central nervous system in our models, we expressed *kRAS*$^{G12D}$ with either DsRED or GFP in *tp53*$^{-/-}$, *tp53*$^{-/-}$ + *TP53*$^{C176F}$ and *tp53*$^{-/-}$ + *TP53*$^{P153\Delta}$ backgrounds (***Figure 5—figure supplement 2***). As previously observed, head tumors were enriched in the *tp53*$^{-/-}$ + *TP53*$^{P153\Delta}$ group and histological analyses revealed that the majority of these tumors had histology consistent with ERMS (***Figure 5—figure supplement 3***). However, we did notice that approximately 20% (5 of 24) displayed histology consistent with medulloblastoma (***Figure 5C–E***). In contrast, the *tp53*$^{-/-}$ group had only one zebrafish (1 of 20 tumors) with a DsRED-positive tumor in the head with histology consistent with medulloblastoma (***Figure 5C–E***). Non-ERMS head tumors highly expressed *sox10* and *gfap* (***Supplementary file 2***) and immunohistochemistry staining confirmed that these brain tumors express high levels of Sox10 and Gfap (***Figure 5F and G***). None of the zebrafish expressing *tp53*$^{-/-}$ + *TP53*$^{C176F}$ initiated head tumors. Finally, by comparing our zebrafish brain tumors with medulloblastoma gene signatures outlined in Pomeroy et al., and Kool et al., we found that our brain tumors expressing *tp53*$^{-/-}$ + *TP53*$^{P153\Delta}$ were consistent with the sonic hedgehog subgroup of medulloblastomas (***Figure 5H***, ***Figure 5—figure supplement 4***; ***Kool et al., 2014***; ***Pomeroy et al., 2002***).

## Expression of *TP53*$^{Y220C}$ predisposes to head ERMS in zebrafish

The location of the deleted proline in *p53*$^{P153\Delta}$ suggested a structural change. We sought to understand the effect of the P153Δ mutation on function by assessing protein structure and stability using in silico modeling. Homology models of p53$^{P153\Delta}$ generated by SWISS-MODEL (***Waterhouse et al., 2018***) indicate that deletion of P153 causes a partial narrowing of a small pocket on the surface of p53$^{WT}$ (***Figure 6A***). P153 is the C-terminal residue of a tri-proline surfaced-exposed loop that retains flexibility and mobility. Conversely, another known p53 mutation, the p53$^{Y220C}$ mutation, causes expansion and deepening of this pocket by forming a cleft bounded by L145, V147, T150-P153, P222, and P223 (***Figure 6A***). We searched the TCGA database for other *TP53* mutants sequentially and structurally close to P153 and identified proline residues P151 and P152 that are also mutated in cancers. Interestingly, mutations in P151 and P152 are present in a subset of tumors overlapping with the Y220C mutation (TCGA Research Network data; http://cancergenome.nih.gov). Previous molecular dynamics simulations of p53$^{Y220X}$ (X = C, H, N, or S) mutations found that the tri-proline loop (P151-153) is mobile and can precipitate a concerted collapse of the pocket, forming a frequently populated closed state and contributing to p53 instability (***Bauer et al., 2020***). Since the binding of small molecules in the pocket increases the stability of p53$^{Y220X}$ mutants, it is not inconceivable that mutations lining this pocket such as the tri-proline loop could be involved in fluctuations that decrease structural stability. Similarly, in our static models, the loss of P153 caused this pocket to be slightly occluded, suggesting further destabilization and abnormal p53 function. (***Figure 6A***). The *TP53*$^{Y220C}$ mutation

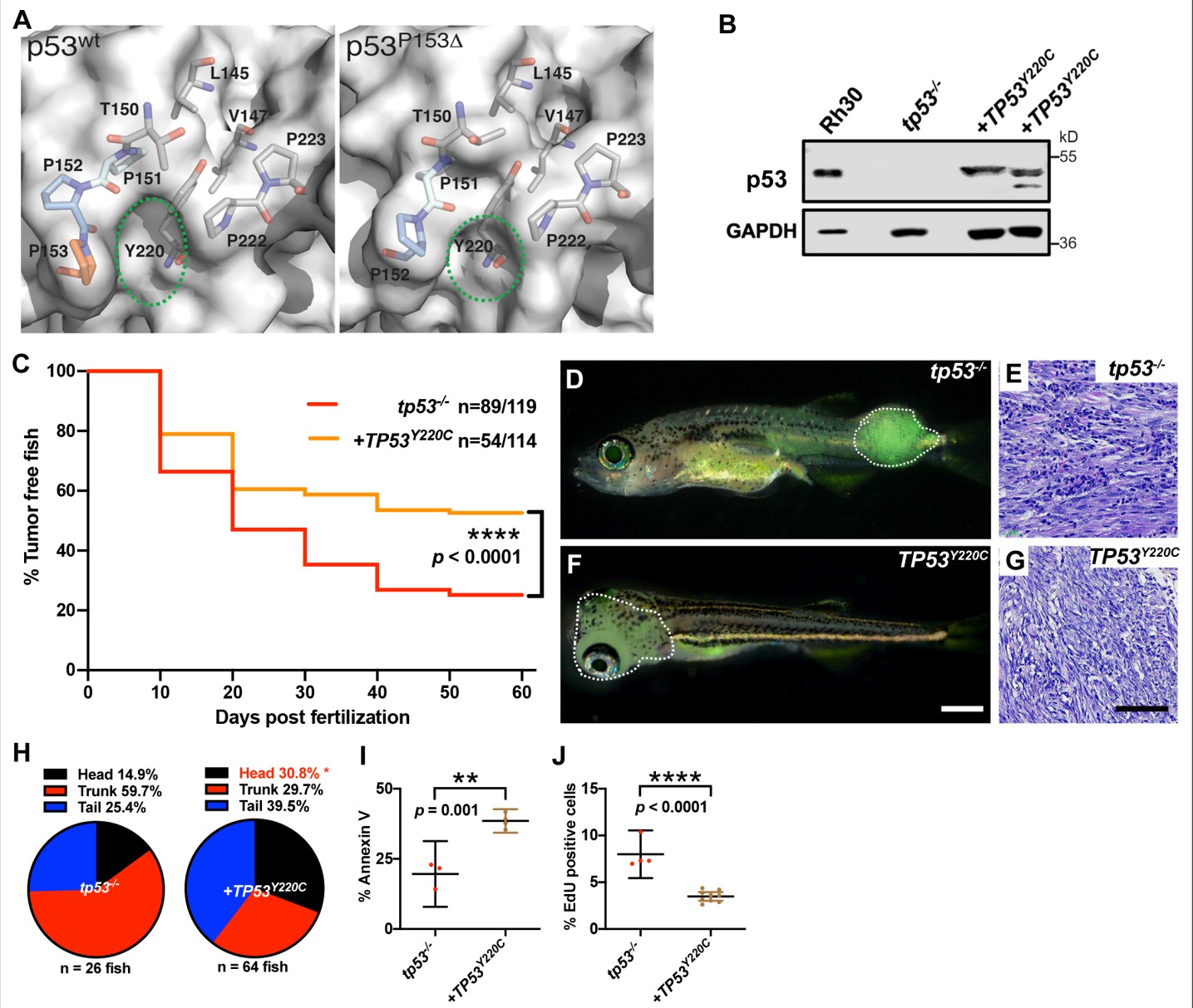

**Figure 6.** *TP53*$^{Y220C}$ predisposes to head embryonal rhabdomyosarcoma (ERMS) in zebrafish. (**A**) Surface representation of p53$^{WT}$ (PDB 2XWR) and p53$^{P153Δ}$ (homology model) showing key residues lining a surface exposed pocket (sticks). The green ovals compare the size and shape of the pocket between the two structures. (**B**) p53 protein expression levels in *tp53*$^{-/-}$ fish tumors with or without *TP53*$^{Y220C}$, with rhabdomyosarcoma (RMS) cell line, Rh30, as a control. (**C**) Kaplan–Meier plot showing tumor initiation in *tp53*$^{-/-}$ fish, with or without *TP53*$^{Y220C}$. (**D, F**) Representative images of *tp53*$^{-/-}$ fish with ERMS tumors, with or without *TP53*$^{Y220C}$ (GFP-positive). Dashed region outlines the tumor. The zebrafish in (**F**) are 35 d. Scale bar in (**F**) 1 mm. (**E, G**) Representative H&E staining of tumors in *tp53*$^{-/-}$ fish, with or without *TP53*$^{Y220C}$. Scale bar = 100 µm. (**H**) Pie chart showing localization of tumors expressed as a percentage found in varying regions of in *tp53*$^{-/-}$ fish with and without *TP53*$^{Y220C}$. Percentage in red indicates a significant difference to *tp53*$^{-/-}$ (p=0.01928, two-tailed two proportions Z-test). (**I**) Quantification of Annexin V staining in tumors of *tp53*$^{-/-}$ fish with or without expression of *TP53*$^{Y220C}$. n = 3–4. (**J**) Quantification of EdU staining in tumors of *tp53*$^{-/-}$ fish with or without expression of *TP53*$^{Y220C}$. n = 4–9.

The online version of this article includes the following figure supplement(s) for figure 6:

**Figure supplement 1.** Schematic of the experimental setup to generate embryonal rhabdomyosarcoma (ERMS) tumors in zebrafish via microinjection of indicated linearized DNA constructs into the one-cell-stage zebrafish embryos generated from in crosses of *tp53*$^{-/-}$ zebrafish.

**Figure supplement 2.** *TP53*$^{Y220C}$ predisposes to head embryonal rhabdomyosarcoma (ERMS) in zebrafish.

**Figure supplement 3.** TP53$^{Y220C}$ predisposes to a round blue cell tumor in zebrafish.

**Figure supplement 4.** Semi-quantitative qPCR analyses comparing expression of known p53 direct regulated genes *baxa, bbc3, cdkn1a, gadd45a,* and *noxa* in primary zebrafish embryonal rhabdomyosarcoma (ERMS) expressing wt *tp53* (blue, n = 3), *tp53*$^{-/-}$ (red, n = 3), and *tp53*$^{-/-}$ + *TP53*$^{Y220C}$ (green, n = 3).

is the ninth most frequent *TP53* missense mutation and is the most common 'conformational' *TP53* mutation in cancer (***Baud et al., 2018***). Interestingly, the *TP53*[Y220C] allele is frequently associated with sarcomas and head and neck carcinomas, and has been reported in patients with osteosarcoma and RMS (***Castresana et al., 1995***; ***Overholtzer et al., 2003***). We therefore hypothesized that loss of P153 via an in-frame deletion might result in similar structural defects as *TP53*[Y220C] by contributing to the overall structural instability (***Wang and Fersht, 2017***). Therefore, we predicted that ERMS tumors expressing *TP53*[Y220C] may phenocopy *TP53*[P153Δ] with increased ERMS in the head region. To test this hypothesis, we generated ERMS that expressed *TP53*[Y220C] with *kRAS*[G12D] and *GFP* in *tp53*[-/-] embryos (***Figure 6—figure supplement 1***). Western blot analyses confirmed that mutant p53[Y220C] protein was expressed (***Figure 6B***). Kaplan–Meier analyses indicated that while *TP53*[Y220C] expression inhibited ERMS initiation (***Figure 6C***, p<0.0001) and decreased the number of primary ERMS per fish (***Figure 6—figure supplement 2A***, p<0.0001), *TP53*[Y220C] expression also led to a significant increase in head ERMS tumors compared to *tp53*[-/-] animals, reminiscent of *tp53*[-/-] animals expressing *TP53*[P153Δ] (***Figure 6D, F and H*** and ***Figure 6—figure supplement 2B***; 30.8% of ERMS; n = 150). We confirmed ERMS pathology on tumor sections (***Figure 6E and G***). Histological analyses confirmed ERMS in 11 of 12 tumor burdened animals, with one tumor displaying characteristics of round blue cell tumor (***Figure 6—figure supplement 3***). We also assessed *TP53*[Y220C]-expressing tumors for effects on proliferation and apoptosis and similar to *tp53*[-/-]; *TP53*[P153Δ] ERMS, expression *TP53*[Y220C] decreased tumor cell proliferation (***Figure 6J***, p<0.0001). However, unlike *TP53*[P153Δ]-expressing ERMS, *TP53*[Y220C] expression led to a significant increase in apoptosis (***Figure 6I***, p=0.001). Lastly, we performed qPCR analyses on bulk tumors to assess whether *TP53*[Y220C] differentially regulated known targets of wild-type p53, including *baxa, cdkn1a, gadd45a, noxa,* and *puma/bbc3*. Compared to *tp53*[+/+] and *tp53*[-/-] tumors, we found that *tp53*[-/-] + *TP53*[Y220C] tumors expressed significantly higher *gadd45a*, and two of three tumors expressed higher *noxa* and *baxa* but not *bbc3* or *cdkn1a/p21* (***Figure 6—figure supplement 4***).

Altogether, our data reveal that both the *TP53*[P153Δ] and *TP53*[Y220C] mutations predispose to head ERMS tumors in zebrafish but differ in their effects on tumor initiation, apoptosis and expression of p53 target genes noxa, baxa, bbc3 and gadd45a (Supplementary***Supplementary file 1*** File 1).

## *kdr* downstream of *TP53*[P153Δ] predisposes to head ERMS in tp53[-/-] zebrafish

In a mouse genetic model, activated hedgehog signaling can initiate head and neck ERMS from a potential bipotent Kdr-positive progenitor cell that has endothelial and myogenic potential (***Drummond et al., 2018***). To test whether head ERMS in zebrafish tumors have molecular features distinct from tumors arising in the trunk, we used RNAseq analyses to compare gene expression between sorted secondary head and trunk tumors generated in tp53[-/-] CG1 syngeneic animals (***Supplementary file 3***). The head ERMS tumor, while having common shared myogenic features, was distinguishable from trunk tumors and expressed several genes that are normally expressed in the head musculature, including *tbx1, dlx3b, dlx4b,* and *ptch2* (***Figure 7A***, ***Supplementary file 3***), suggesting that head ERMS are molecularly different from trunk tumors. Mutant *TP53* has been previously shown to activate *KDR* expression in breast cancer cell lines (***Pfister et al., 2015***) and high *KDR* expression in a subset of osteosarcoma tumors is associated with poor outcome (***Negri et al., 2019***). Clinical data from our *TP53*[P153Δ] osteosarcoma patient revealed *KDR* amplification. From the patient-derived xenograft, we confirmed KDR expression using IHC analysis and using western blot found that while commonly used *TP53* null osteosarcoma SaOS2 cells do not express KDR, our *TP53*[P153Δ] osteosarcoma PDX robustly expressed KDR (***Figure 7B and C***). Therefore, we hypothesized that loss of *kdr* expression in *tp53*[-/-] zebrafish co-expressing *rag2:kRAS*[G12D] and *rag2:TP53*[P153Δ] would inhibit ERMS initiation in the head. To test this, we injected *rag2:kRAS*[G12D], *rag2:TP53*[P153Δ], *rag2:GFP* together with Cas9 protein and guide RNAs (gRNAs) targeting either *kdr* (catalytic domain) or *mitfa* (control) in *tp53*[-/-] embryos (***Figure 7D and E***). We expected that zebrafish would tolerate mosaic loss of *kdr* due to the presence of *kdrl* (***Bussmann et al., 2008***), while *mitfa* would be required for melanophore specification, but not viability in zebrafish (***Lister et al., 2001***). We first assessed tumor initiation and found that mosaic targeting of *kdr* resulted in an overall reduction in tumor initiation compared to *mitfa* (***Figure 7F***). Importantly, *kdr* CRISPR/Cas9 resulted in a significant reduction in head ERMS tumors (***Figure 7G–I***), while targeting *mitfa* did not suppress head ERMS initiation. We confirmed efficient CRISPR/Cas9 targeting using PCR, identifying deletions in exons 12 and 13, which contain the *kdr* catalytic domain (***Figure 7J***, ***Figure 7—figure supplement 1***). Altogether, our data supports a role for mutant *TP53*[P153Δ]

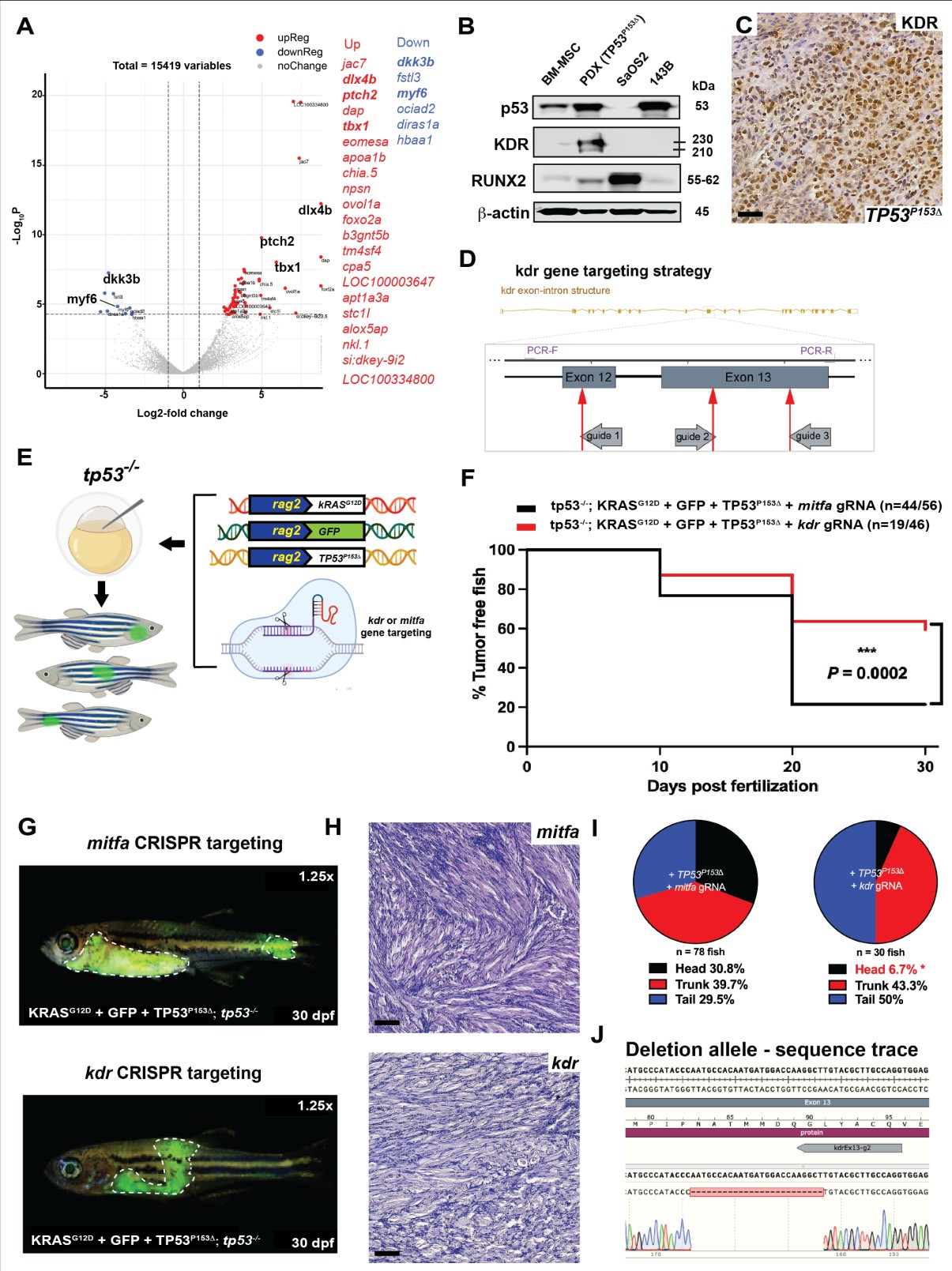

**Figure 7.** *kdr* downstream of *TP53^P153Δ^* predisposes to head embryonal rhabdomyosarcoma (ERMS) in tp53^-/-^ zebrafish. (**A**) Volcano plot comparing head to trunk ERMS tumors. Genes with adj. p-value<0.01 and fold-change >2 are colored in red (upregulated in head tumor) or blue (downregulated in head tumor). Table adjacent the plot shows the top differentially expressed genes. (**B**) Western blot showing p53, KDR, and RUNX2 protein expression in bone marrow mesenchymal cells, the osteosarcoma PDX expressing *TP53^P153Δ^*, and osteosarcoma cell lines SaOS2 and 143B. (**C**) Representative KDR

*Figure 7 continued on next page*

*Figure 7 continued*

IHC staining of osteosarcoma PDX expressing p53$^{P153Δ}$. Scale bar = 60 μm. (**E**) Schematic demonstrating microinjection into the one-cell-stage zebrafish embryos with the indicated constructs to generate GFP-expressing tumors with either *kdr* or *mitfa* mosaically deleted in vivo. (**F**) Kaplan–Meier plot showing ERMS tumor initiation in *tp53$^{-/-}$* fish expression *TP53$^{P153Δ}$* with either *mitfa* or *kdr* ablation. (**H**) Representative H&E staining of tumors in *tp53$^{-/-}$* fish or *tp53$^{-/-}$* fish expressing *TP53$^{P153Δ}$* with either *mitfa* or *kdr* ablation. Scale bar = 60 μm (**G**) Representative images of tumor localization in *tp53$^{-/-}$* fish with expression of *TP53$^{P153Δ}$* with either *mitfa* or *kdr* deletion. Fish in the panels are 30 days old. (**I**) Pie chart showing the percentage of tumors found in varying regions of *tp53$^{-/-}$* fish with expression of *TP53$^{P153Δ}$* with either *mitfa* or *kdr* deletion, showing significant differences in the head region. *p=0.0088 (two-tailed two proportions Z-test). (**J**) Example of sequencing from an ERMS tumor with a deletion in a portion of *kdr* exon 13. Five of five of ERMS tumors analyzed have deletions in *kdr*.

The online version of this article includes the following figure supplement(s) for figure 7:

**Figure supplement 1.** kdr downstream of TP53$^{P153Δ}$ predisposes to head embryonal rhabdomyosarcoma (ERMS) in tp53-/- zebrafish.

functioning in part through *kdr* to influence the formation of ERMS tumors in head musculature. However, it is important to state that our mosaic analyses are unable to distinguish whether *TP53$^{P153Δ}$* regulation of *kdr* expression is required cell autonomously or in the tumor vasculature.

## Discussion

The *TP53* tumor-suppressor gene is mutated in >40% of human tumors, and patients with Li–Fraumeni syndrome with germline mutations are predisposed to a spectrum of tumors that includes several lethal childhood sarcomas, such as rhabdomyosarcoma and osteosarcoma (***Gröbner et al., 2018***; ***Guha and Malkin, 2017***). Here, we addressed three poorly understood aspects of *TP53* function in ERMS, a devastating pediatric malignancy of the muscle. First, we found that the *tp53* pathway is a major suppressor of tumor initiation in RAS-driven ERMS. Second, we establish that human *TP53* can complement zebrafish *tp53* function. Third, we utilized our robust zebrafish ERMS model to assign function to three mutations whose effects in ERMS were previously unknown. We defined *TP53$^{C176F}$* as a hypomorphic allele and *TP53$^{P153Δ}$* as pathogenic with gain-of-function activity. We found that the structurally related mutants *p53$^{P153Δ}$* and *p53$^{Y220C}$* predispose to head musculature ERMS (***Supplementary file 1***). Finally, p53$^{P153Δ}$ also predisposes to hedgehog driven medulloblastomas.

Patients with mutant RAS-driven ERMS have high-risk disease and are a challenge to treat (***Chen, 2013***; ***Shern et al., 2014***). Murine RAS mutant ERMS models require loss of *Trp53* and/or *p16* (***Kashi et al., 2015***), making the zebrafish ERMS model more similar to human in that RAS activation is sufficient to drive tumor formation. However, a maximum of 40% of wild-type *kRAS$^{G12D}$*-expressing zebrafish initiate tumors by 50 days of life (***Langenau et al., 2007***). Interestingly, both human and zebrafish ERMS with wild-type *TP53* display pathway inhibition with *MDM2* amplification and/or over-expression (***Chen, 2013***; ***Langenau et al., 2007***; ***Seki et al., 2015***; ***Shern et al., 2014***), suggesting a role for p53 pathway suppression during ERMS initiation. Additionally, genes such as *TWIST1* have been shown to promote tumor initiation in sarcomas by inhibiting p53 expression (***Piccinin et al., 2012***). Of note, mutant kRAS-induced ERMS formation increases to approximately 70% in *tp53$^{M214K/M214K}$* mutant zebrafish (***Berghmans et al., 2005***; ***Langenau et al., 2007***), suggesting that *tp53* plays an important role in suppressing ERMS initiation.

The *tp53$^{-/-}$* complete loss-of-function mutant zebrafish line (***Ignatius et al., 2018***) spontaneously forms a broader tumor spectrum than *tp53$^{M214K/M214K}$*, including MPNSTs, angiosarcomas, NK-cell leukemias, as well as germ cell tumors (***Berghmans et al., 2005***; ***Ignatius et al., 2018***), suggesting the possibility of differential effects in RAS-driven ERMS. Thus, our study tested the role for loss of *tp53* in ERMS initiation by generating *kRAS$^{G12D}$*-induced ERMS in the *tp53$^{-/-}$* background. We found that >97% of animals form ERMS, revealing that *tp53* pathway suppression is required for full tumor penetrance. Moreover, when compared to *tp53* wild-type, *tp53$^{-/-}$* animals display significantly more tumors per animal, increased tumor cell proliferation, and minor effects on tumor cell apoptosis. These effects on apoptosis can be enhanced by wild-type human *TP53* or zebrafish *tp53* expression specifically in ERMS tumor cells using the zebrafish *rag2* promoter. Our previous work showed that *tp53* loss specifically enhanced invasion and metastasis in vivo (***Ignatius et al., 2018***), suggesting that while *TP53* is shown to suppress tumor progression by multiple mechanisms depending on tumor type, loss of *TP53* in human ERMS may influence clinical outcomes through a combination of increased tumor cell proliferation, invasion, and metastasis (***Ignatius et al., 2018***).

A second important finding is that human *TP53* can complement wild-type *tp53* function in zebrafish ERMS, which allows for direct study of different human variants in vivo. Overall human and zebrafish *TP53* show a high level of similarity (56% identity), with the DNA-binding domain displaying especially high homology with 72% amino acid identity (79% positive substitutions) and no gaps in the sequence (*Figure 3—figure supplements 1 and 2*). We found that both wild-type human and zebrafish *Tp53* suppresses ERMS initiation and decreases proliferation in the *tp53*[-/-] background. However, we observed a significant increase in apoptotic cells only with overexpression, suggesting that higher levels of p53 induction may be required to induce apoptosis in a wild-type background. Our co-expression approach has been previously used to study multiple aspects of tumorigenesis in ERMS, T-cell acute lymphoblastic leukemia (T-ALL), melanoma, liver cancer, and neuroblastoma (*Blackburn et al., 2014*; *Langenau et al., 2007*; *Lobbardi et al., 2017*; *White et al., 2013*), and results in protein expression comparable to or lower than p53 in human Rh30 RMS cells that endogenously express mutant p53[R273C] (*Gibson et al., 1998*). Co-expression has been an effective method to assess the role for multiple signaling pathways in ERMS, including Notch, canonical and non-canonical Wnt, Myf5, and MyoD, with pathway validation in human cells using in vitro and in vivo methods (*Chen et al., 2014b*; *Hayes et al., 2018*; *Ignatius et al., 2017*; *Tenente et al., 2017*).

Having established that wild-type human *TP53* functions similarly to zebrafish *tp53* in ERMS, we are now able to assess patient variants identified in sarcomas and define their roles or activity during tumorigenesis (*Supplementary file 1*). A *TP53*[C176F] mutation was identified in a patient with ERMS that displayed functional loss of the second *TP53* allele (*Chen, 2013*). A second *TP53*[P153Δ] mutation was found as a germline variant in a patient with osteosarcoma. Finally, the *TP53*[Y220C] mutation is commonly found in sarcomas, including RMS and osteosarcoma (*Castresana et al., 1995*; *Overholtzer et al., 2003*). Using our in vivo assays, we were able to define *TP53*[C176F] as a hypomorph with respect to ERMS initiation that retains the ability to induce apoptosis. These findings are consistent with studies showing that *TP53*[C176F] does retain some aspects of wild-type p53 function, can form tetramers with wild-type p53, and can differentially induce *TP53* target gene transcription (*Hoffman-Luca et al., 2015*; *Kato et al., 2003*). Compounds like ZMC1 that stabilize mutant protein to enhance p53-mediated tumor cell apoptosis and decrease growth (*Blanden et al., 2020*; *Blanden et al., 2015*; *Yu et al., 2012*) selectively increased ERMS apoptosis in *tp53*[-/-] + *TP53*[C176F] but not in *tp53*[-/-] tumors, further highlighting our assay as an effective in vivo drug efficacy screening tool. Altogether, our data shows that *TP53*[C176F] is likely hypomorphic and chemical stabilization of the resulting mutant protein could be an effective therapeutic strategy for patients with this allele.

In contrast to *TP53*[C176F] or *TP53*[Y220C], the *TP53*[P153Δ] variant is rare, with only one other known patient reported (*Michalarea et al., 2014*). This patient was diagnosed with multiple tumors over several decades, including bilateral breast cancer, malignant fibrous histocytoma, and an EGFR mutant lung adenocarcinoma. The patient's mother, maternal aunts, and maternal grandmother all experienced early-onset cancers, meeting the criteria for a Li–Fraumeni diagnosis (*Michalarea et al., 2014*). The specific Proline 153 residue, while in a region that is highly conserved across mammalian species (*Figure 3—figure supplement 2*), is not conserved in mice or zebrafish, creating a challenge for modeling endogenous p53[153] expression in vivo. Due to its rarity in the literature and lack of animal models, genetic testing done through Invitae, a medical genetic testing provider, could not assign pathogenicity to *TP53*[P153Δ]; however, the family experience strongly suggested that this is a Li–Fraumeni variant. Clinically, the patient's osteosarcoma was extremely aggressive and refractory to multiple anticancer agents. Using our in vivo model, we found that *TP53*[P153Δ] predisposes to head ERMS that are relatively resistant to apoptosis. Currently, no other gene or pathway has been identified in the zebrafish model that predisposes to head tumors, with both *TP53*[P153Δ] and *TP53*[Y220C] expression in zebrafish leading to an increase in head ERMS. This reveals two *TP53* variants that may have shared gain-of-function effects in ERMS with respect to the initiation of head tumors. Unexpectedly, approximately 20% of *tp53*[-/-] zebrafish expressing *TP53*[P153Δ] also develop medulloblastoma-like tumors with gene signatures consistent with the sonic hedgehog subgroup. Brain tumors have not been observed in *rag2:kRAS*[G12D] –induced ERMS model, and we did not identify any brain tumors in our *TP53*[Y220C] or *TP53*[C176F] experimental groups. We did identify however one medulloblastoma-like tumor in *tp53*[-/-] zebrafish. Given the rapid onset of ERMS is our model, it is possible that medulloblastoma requires more time to initiate and that removal of ERMS burdened animals from the experiment before 30 days precludes study of additional tumor types in *tp53*[-/-] animals. Tumors in the *tp53*[-/-] + *TP53*[C176F] group

grew slowly and experimental animals were followed for up to 90 days, yet no medulloblastoma tumors were observed, suggesting that hypomorphic $TP53^{P153\Delta}$ activity is likely not sufficient to initiate medulloblastoma tumors, in addition to head and neck ERMS.

Head and neck ERMS represent a significant proportion of all ERMS diagnoses, with the basis of regional predispositions not fully understood. Expression of activated Smoothened protein in mice under the control of the Ap2 promoter results primarily in head and neck ERMS that arise from Kdr-positive endothelial cells (*Hatley et al., 2012*). Smoothened is a key component of the hedgehog signaling pathway that is commonly activated in ERMS (*Drummond et al., 2018*; *Satheesha et al., 2016*). Modeling in mice indicates that the cell of origin plays a major role in initiation, given that activated Smoothened under the control of a more ubiquitously expressed promoter leads to ERMS tumors in other skeletal muscle populations (*Mao et al., 2006*). Importantly, a recent analysis of *TP53* in RMS found a higher proportion of head and neck RMS tumors have *TP53* mutations (*Shern et al., 2021*). Our data suggests that differential features of $TP53^{P153\Delta}$ and $TP53^{Y220C}$ may predispose patients to head ERMS via regulation of *kdr* expression, providing additional mechanistic insights into anatomical differences in sarcoma initiation. Therefore, certain *TP53* variants may predispose individuals to head and neck rhabdomyosarcoma tumors; however, the cell of origin for zebrafish head ERMS remains to be identified, as well as whether mutant *TP53* interacts with hedgehog pathway signaling in head ERMS and medulloblastoma.

Finally, we would like to state the limitation of our zebrafish ERMS model. In Li–Fraumeni patients, *TP53* mutations are germline with somatic modifier mutations occurring secondarily. In our experiments, while our starting point is a tp53⁻/⁻ null background, $kRAS^{G12D}$ and the human *TP53* variant allele are introduced at the same time under the control of the *rag2* promoter. A second limitation is that effects of mutant *TP53* in the tumor microenvironment are missed in our assays. However, our assay is rapid, given that the generation of zebrafish strains with germline knock-in of the equivalent mutations takes an average of 2 years for full assessment. In mice, only the most common *Tp53* variants have been fully characterized, and prioritizing which *TP53* mutants to model is a challenge given cost and time restraints. Our mosaic assay bridges a gap, especially for sarcomas where the majority of *TP53* mutations remain uncharacterized. Our finding that expression of three different patient *TP53* variants leads to very different effects on tumor formation highlights the importance of in vivo precision modeling, with our model promising to help further define patient-specific p53 biology. In summary, we highlight that the zebrafish ERMS model can be effective in defining multiple aspects of p53 tumor suppressor function and delineating a spectrum of null, partial loss- and gain-of-function mutational effects in vivo.

## Materials and methods
### Animals
Animal studies were approved by the UT Health San Antonio Institutional Animal Care and Use Committee (IACUC) under protocol #20150015AR (mice) and 20170101AR (zebrafish). Zebrafish strains used in this work include AB wild type, AB/tp53⁻/⁻, CG1 wild type, CG1 /tp53⁻/⁻, Tg casper; *myf5:GFP*. Zebrafish were housed in a facility on a continuous flow system (Aquarius) with temperature-regulated water (~28.5°C) and a 14 hr light–10 hr dark cycle. Adult fish were fed twice daily with brine shrimp, supplemented with solid food (Gemma). Larval fish were kept off the continuous flow system until 15 days post-fertilization and supplemented with a paramecium/algae culture before transferred online.

### Generation of zebrafish rhabdomyosarcoma tumors
*rag2:kRASG12D* and *rag2:DsRed* plasmids were linearized with XhoI, followed by phenol–chloroform extraction and ethanol precipitation. The purified DNA was resuspended in nuclease-free water (AM9916, Thermo Fisher) and injected into embryos at the one-cell stage of development (40 ng/µl of *rag2:kRASG12D* and 20 ng/µl of *rag2:DsRed*). For expression of zebrafish *tp53* or human *TP53* in ERMS tumors in zebrafish, XhoI-linearized *rag2:kRASG12D* and *rag2:DsRed* plasmids were injected along with Xmn1-linearized *rag2:TP53* (human) or *rag2:tp53* (zebrafish) constructs. For expression of mutant human *TP53* experiments, tumors were generated by injecting XhoI-linearized *rag2:kRASG12D* and *rag2:DsRed* or *rag2:GFP* along with Xmn1-linearized *rag2:TP53^{C176F}*, or *TP53^{P153Δ}*, or *TP53^{Y220C}* (35 ng/

μl of *rag2:kRASG12D,* 15 ng/μl *rag2:TP53* wild-type or mutant *TP53* and 10 ng/μl of *rag2:DsRed*) into the one-cell stage of zebrafish embryos <1 hr post-fertilization (*Ignatius et al., 2012*; *Langenau et al., 2007*). For testing the role of *kdr* on head ERMS initiation *TP53* experiments, tumors were generated by injecting XhoI-linearized *rag2:kRASG12D, rag2:GFP* and *TP53^{P153Δ}* along with a CAS9 protein + guide RNA complex (35 ng/μl of *rag2:kRASG12D,* 15 ng/μl *rag2:TP53^{P153Δ}*, 10 ng/μl of *rag2:DsRed* and 200 ng/μl CAS9 + 100 ng/μl gRNA1 *kdr* + 100 ng/μl gRNA2 *kdr* + 100 ng/μl gRNA3 *kdr* or 100 ng/μl *mitf*) into the one-cell stage of zebrafish embryos <1 hr post-fertilization. Animals were allocated into experimental groups based on which injection cocktail they received during microinjection. Animals were monitored for tumor onset beginning at 10 d post-fertilization by scoring for DsRed or GFP fluorescence under an Olympus MVX10 stereomicroscope with an X-Cite series 120Q fluorescence illuminator. Scoring for tumor initiation was conducted for 60 days. The tumor onset was visualized by Kaplan–Meier plots, and the significance was analyzed by log-rank (Mantel–Cox) test for each two groups at a time. Sample size was determined taking into account the observed tumor initiation rates from previous studies using the zebrafish ERMS model (*Langenau et al., 2008*; *Tenente et al., 2017*).

## Tumor size (ratio) measurements

Tumor size was measured using FIJI image analysis software (https://fiji.sc/) by comparing area occupied by the tumor region to the total body area of the fish. Specifically, all tumor-burdened fish were imaged in similar lateral position under both bright-field (white) and fluorescent light. The final representative images were generated by superimposing bright-field and fluorescence images using Adobe Photoshop. Significance was assessed by Student's *t*-test.

## EdU staining

5-ethynyl-2'-deoxyuridine (EdU, Molecular Probes, Life Technologies) was dissolved in DMSO to make a 10 mM stock solution. This stock solution was further diluted 50 times using PBS to 200 μM, and 0.15 μl and 0.3 μl were injected into juvenile and older zebrafish, respectively. After 6 hr EdU treatment, tumor fish were euthanized with an overdose of MS-222 (Tricaine) and fixed in 4% paraformaldehyde (PFA) at 4°C overnight. Then, the fixed samples were soaked using 25% sucrose (in PBS) at 4°C overnight. Finally, the samples were embedded in OCT medium and the medium was allowed to solidify on dry ice. Tissue was sectioned using a Leica CM1510 S Cryostat and tissue sections were placed on plus gold microscope slide (Fisherbrand). The slides were post-fixed in 4% PFA for 15 min and permeabilized in 0.5% Trition x-100 in BPS for 20 min. Then, freshly made Click-iT Plus reaction cocktail was added to each slide and allowed to incubate for 30 min. A small amount of Vectashield mounting medium with DAPI was added to the slide and covered with a coverslip. The EdU Click-iT Plus EdU Alexa Fluor 647 Imaging kit (Molecular Probes, Life Technologies) was used for EdU staining. Tissue sections (one section per tumor) were subsequently analyzed using an Olympus FV3000 confocal microscope and percentage of Edu-positive determined by counting the total number of DAPI-positive nuclei in tumor section. Significance was assessed by Student's *t*-test. This assay was completed using at least n ≥ 3 biological replicates (multiple primary tumors or transplanted syngeneic zebrafish tumors).

## qPCR analysis

Tumors were extracted from euthanized fish and homogenized inside a 1.5 ml Eppendorf tube using a pestle connected to a Pellet Pestle Cordless Motor (DWK Life Sciences Kimble Kontes). Tumor RNA was extracted using NEB's Monarch RNA mini prep kit. cDNA was synthesized from the extracted RNA using Invitrogen's First Strand Synthesis cDNA kit. Following cDNA synthesis, real-time qPCR was performed in 384-well plates using Applied Biosystem's Sybr Green qPCR MasterMix (Comparative $C_T$ ($\Delta\Delta C_T$) method). Results were compiled and analyzed using the QuantStudio 7 Flex system (Applied Biosciences). PCR primers are provided in *Supplementary file 4*. All assays were completed using technical replicates, with at least three tumors tested per experimental group.

## Phospho-histone H3 staining

Fish were fixed in 4% PFA at 4°C overnight. The fixed fish were subsequently soaked in 25% sucrose overnight and then embedded in OCT medium before being sectioned at 10 μm with a Leica CM1510 S cryostat. After being washed three times in PBST (0.1% Triton X-100 and 0.1% Tween 20 in PBS),

the sections were incubated in blocking solution (2% horse serum, 10% FBS, 0.1% Triton X-100, 0.1% Tween 20, 10% DMSO in PBS) for 60 min. The sections were then incubated with rabbit anti-phospho-histone H3 (Ser10) primary antibody (1:500 dilution) at 4°C overnight. The following day, sections were washed three times in PBST and incubated with Alexa Fluor 647 conjugated anti-rabbit secondary antibody at room temperature for 2 hr. Vectashield mounting medium with DAPI was added to the slide and then a coverslip was placed over the sample. The slides were dried in the dark and sealed by nail polish. Sections were imaged using an Olympus FV3000 confocal microscope. Significance was assessed by Student's *t*-test. This assay was completed using at least three biological replicates and counting total number of positive cells per a single confocal image taken at ×200 magnification of a tumor section (n ≥ 3 primary tumors).

### Annexin V-FITC/PI staining

Fish were euthanized and tumor was isolated, following which the tumor was homogenized in 0.9× PBS + 5% FBS manually using a razor blade and made into single suspensions using 45 micron filters. Tumor cells were washed with 0.9× PBS + 5% FBS and resuspended in the binding buffer containing Annexin V-FITC and propidium iodide for 15 min in the dark at room temperature. Then the cells were detected by flow cytometry (FCM, FACS Canto, BD, CA). Significance was assessed by Student's *t*-test. This assay was completed using at least three biological replicates (n ≥ 3).

### Histology and immunohistochemistry

Euthanized zebrafish were fixed in 4% PFA overnight at 4°C. Embedding, sectioning, and immunohistochemical analysis of zebrafish sections were performed as previously described (*Chen et al., 2014b*; *Ignatius et al., 2012*). H&E staining was performed at the Greehey CCRI histology core. Slides were imaged using a Motic EasyScan Pro slide scanner. Pathology review and staging were completed by board-certified pathologists (EYC and ARG).

### Cloning *TP53* wild-type and mutants constructs

Wild-type *TP53* from both human (Addgene plasmid #69003) or zebrafish (3-day-old embryos cDNA) were amplified by PCR and cloned into pENTR-D-TOPO vector, which was verified by DNA sequencing. *rag2:TP53* (human) or *rag2:tp53* (zebrafish) plasmids were generated by one-step Gateway reaction between a Gateway-compatible plasmid with the zebrafish *rag2* promoter flanked by *attR* sites and the respective pENTR-D-TOPO plasmid. *TP53^{C176F}*, *TP53^{P153Δ}*, or *TP53^{Y220C}* fragments were constructed by amplifying human *TP53* as two separate fragments with the respective mutations, with the 3′ end of the first fragment possessing 60 bp of homology with the 5′ end of the second fragment. These two fragments were purified from a 1% agarose gel using a Macherey–Nagel purification kit and spliced together using overlap extension PCR with Phusion high fidelity DNA polymerase (*Szymczak-Workman et al., 2012*). The entire spliced fragment was then blunt-ligated into a pENTR-D-TOPO vector. The *TP53* insert was sequenced after which it was cloned into a Gateway-compatible plasmid with the zebrafish *rag2* promoter flanked by *attR* sites using a one-step Gateway reaction using Gateway LR Clonase Enzyme mix. All other PCR amplification was carried out using Q5 high-fidelity DNA polymerase.

### Western blot analyses

Western blot analysis on fish tumors was performed by first extracting tumor from fish and homogenizing tumor cells suspended in SDS lysis buffer inside a 1.5 ml Eppendorf tube using a pestle connected to a cordless motor. The total protein concentration for each lysate solution was normalized with a BCA assay kit (Thermo Fisher Scientific, Carlsbad, CA). Then, 40 µg of total protein was run on a 10% SDS/PAGE gel. The protein transferred membrane were blocked using 5% fat-free milk in TBST, followed by incubation in the appropriate antibody. The list of antibodies and where they were obtained from is provided in *Supplementary file 5*.

### Identifying *TP53* mutation status in osteosarcoma PDX sample

DNA sample from osteosarcoma PDX was isolated using the QIAGEN Puregene Core Kit A. PDX sample was first homogenized in lysis buffer, followed by heating at 65°C for 30 min. Thereafter, RNAse A was added to the sample and incubated for 30 min at 37°C. Protein precipitation buffer

was used to precipitate protein, and the sample was vortexed and then centrifuged at 15,000 RPM for 3 min. The supernatant was then removed, and isopropanol was added to precipitate genomic DNA, following which the sample was centrifuged at 15,000 RPM for 2 min at 4°C. The supernatant was then drained, and the DNA pellet was washed with 70% ethanol, followed by centrifugation at 15,000 RPM for 1 min. The DNA pellet was then resuspended in DNA hydration solution, incubated at 65°C for 1 hr to dissolve DNA, and then incubated at 22°C for 1 hr. This precipitated DNA was used as a template in a PCR reaction using *TP53*-specific primers flanking the A159 and P153 residues. The PCR amplicon was ligated into a pCR4-TOPO blunt cloning vector, and the ligation mix was transformed into DH5α chemically competent cells. Plasmid DNA was isolated from transformed colonies and sequenced using M13 primers.

## ZMC1 treatment

Tumor-burdened fish were incubated in fish water containing ZMC1 at 70 nM concentration with another batch of tumor-bound fish incubated in 0.1% v/v DMSO (diluted in fish water) as control (*Supplementary file 6*). Fish were imaged every week on an Olympus MVX10 stereomicroscope with an X-Cite series 120Q fluorescence illuminator, and size of tumors was measured from the acquired images using FIJI software.

## FACS sorting

FACSorting was completed essentially as previously described (*Chen et al., 2014a*; *Ignatius et al., 2012*; *Ignatius et al., 2018*). Tumor-burdened syngeneic CG1 zebrafish with secondary head tumors expressing *kRAS^{G12D}; TP53^{P153Δ}; GFP* were euthanized and tumor isolated and made into single-cell suspensions in 0.9× PBS containing 5% FBS (*Chen et al., 2014a*; *Ignatius et al., 2012*; *Ignatius et al., 2018*). Live tumor cells were stained with DAPI to exclude dead cells and sorted twice using a Laser BD FACSAria II Cell Sorter. Sort purity and viability were assessed after two rounds of sorting, exceeding 85% respectively. GFP+ tumor cells were isolated by FACS from secondary tumor transplanted fish, cells sorted and fixed in RNAlater and/or Trizol and RNA isolated for RNA sequencing.

## RNAseq and bioinformatic analyses

Total RNA was isolated from the six zebrafish tumors using NEB's Monarch RNA mini prep kit (NEB Inc, MA) according to the manufacturer's instructions. DNase (Thermo Scientific, Rockford, IL, Cat# EN0521) was added to the first wash solution at 10 μg/70 μl and incubated for 15 min at room temperature to remove genomic DNA contamination. The quality of RNA samples was analyzed with a Bioanalyzer (Agilent 2100 Bioanalyzer, Agilent Technologies, Santa Clara, CA) by the GCCRI Genome Sequencing Facility. Samples with RNA Integrity Number (RIN) ≥ 7 were used for RNA sequencing library construction using TruSeq Stranded mRNA Library Prep kit according to the manufacturer's protocol (Cat# RS122-2002; Illumina, Inc). Samples were sequenced in the Illumina HiSeq 3000 (Illumina, Inc) using a 50 bp single-read sequencing protocol. All sequence reads were aligned to the UCSC zebrafish genome build danRer11 using TopHat2 and expression quantification with HTSeq-count (*Anders et al., 2015*) to obtain the read counts per gene in all samples. The RNAseq data was deposited on GEO (GSE213869). Differential expression analysis was performed using R package DESeq (*Anders and Huber, 2010*). Statistically and biologically significant DE genes (DEGs) were defined by applying the following stipulations: adjusted p-values<0.01 and fold change ≥2. Gene Ontology enrichment from the DEG list was performed using TopGO package (*Alexa et al., 2006*), and significantly enriched functions were selected based on classic Fisher's exact test p-value<0.05. Gene set enrichment analysis (GSEA) was performed using standard alone algorithm (http://www.gsea-msigdb.org/gsea/index.jsp, *Subramanian et al., 2005*) or R package (R/fgsea, *Korotkevich et al., 2019*). Differential expression was visualized, such as the volcano plot, using R (https://www.r-project.org).

## Homology modeling

The p53^{P153Δ} mutant was modeled using the SWISS-MODEL (*Schwede et al., 2003*) homology modeling server. The mutant p53 sequence and the p53^{WT} crystal DNA-binding domain crystal structure 2XWR (*Natan et al., 2011*) were used as input files. The homology models are built, scored, and selected using statistical potentials of mean force scoring methods. Side-chain rotamers for non-conserved

residues are selected through a local energy minimization function, and the final models are subjected to global energy minimization using the CHARM22/CMAP forcefield (*Waterhouse et al., 2018*).

## Strategy and quantification of mosaic ablation of *kdr* and *mitfa* in zebrafish ERMS tumors using CRISPR/Cas9 reagents

Using Benchling, three high-scoring CRISPR guides (with minimal off-target scores) were selected to target DNA sequences in exons 12 and 13 of *kdr* that encode parts of two Ig repeats in the extracellular domain (aa 535–647; InterPro domain IPR003599). Efficient nuclease activity of guides 1 and 3 would result in a 334 bp deletion, leading to a predicted frameshift and premature termination of the polypeptide sequence. When this larger deletion does not result from synchronous CRIPSR nuclease activity of guides 1 and 3, similar consequences are likely to result due to independent indels generated from nuclease activity and NHEJ repair at each of the three individual target sites of guides 1, 2, and 3. This region being upstream of the transmembrane domain (aa 773–795), we predict that indels or deletions leading to frameshift mutations would, at best, produce a partial extracellular domain containing polypeptide incapable of transducing signals into the cell.

| Guide | Target sequence (PAM) | Benchling (**Hsu et al., 2013**) off-target score |
|-------|----------------------|---------------------------------------------------|
| g1 | GGTAGCGATGCACCTGTATA (GGG) | 95.9 |
| g2 | CTATAACTTGCGCTGGTATC (GGG) | 96.4 |
| g3 | CCTGGCAAGCGTACAAGCCT (TGG) | 98.4 |

Using PCR and TBE-PAGE, the presence of deletions was visualized by small PCR amplicons corresponding to a loss of 334 bp in a 641 bp wild-type PCR product (deletion mutant amplicon = 212 bp). In addition, presence of heteroduplex products was observed via PAGE, indicating the presence of indels. Indels were confirmed by cloning PCR products and sequencing individual clones. Shown in *Figure 7—figure supplement 1D* are indels +21 bp, –7 bp at the g1 site, 7 bp deletion at the g2 site, and 25 bp deletion at the g3 site. The effectiveness of the three sgRNA/CRISPR guides by way of the spectrum of mutations they cause suggests that despite the mosaicism of the mutations in sgRNA-Cas9-injected embryos, tumors that arise harbor a relatively high-frequency loss-of-function mutations in the *kdr* gene in all tumors. *mitfa* was selected as a control gene because loss-of-function mutations can be easily screened by looking for the loss of pigmentation in developing larvae using brightfield microscopy. In addition to not having an effect on embryo survival and development, it obviates the need to obtain sequence data to confirm the presence of loss-of-function mutations in the embryos.

## Acknowledgements

This project has been funded with federal funds from NIH grants MI and PH (R00CA175184), Cancer Prevention & Research Institute of Texas (CPRIT)-funded Scholar grant to MI (RR160062). JC was supported by Wenzhou Medical University young scientist training program (2019) and the research program (KYYW202203). DSL was supported by the St. Baldricks Foundation and the Welch Foundation. DSL and MI are each recipients of the Max and Minnie Tomerlin Voelcker Fund Young Investigator Awards. KB is a T32 and TL1 fellow (T32CA148724, TL1TR002647). NH was supported by the Greehey CCRI Graduate Student Fellowship and the Cancer Prevention & Research Institute of Texas (CPRIT)-funded Research Training Award (RP 170345). AL was supported by the Cancer Prevention & Research Institute of Texas (CPRIT)-funded Research Training Award (RP 170345) and by a Hyundai Hope On Wheels Young Investigator Grant. RNA sequencing data was generated in the GCCRI Genome Sequencing Facility, which is supported by GCCRI, NIH-NCI P30 CA054174 (NCI Cancer Center Support Grant UT Health San Antonio), NIH Shared Instrument grant 1S10OD030311-01, and CPRIT Core Facility Award RP160732.

## Additional information

### Funding

| Funder | Grant reference number | Author |
|---|---|---|
| National Institutes of Health | R00CA175184 | Peter Houghton<br>Myron S Ignatius |
| Cancer Prevention and Research Institute of Texas | Scholar Grant RR160062 | Myron S Ignatius |
| Wenzhou Medical University | Young Scientist Training Program (2019) | Jiangfei Chen |
| Wenzhou Medical University | KYYW202203 | Jiangfei Chen |
| St. Baldrick's Foundation | | David S Libich |
| Welch Foundation | | David S Libich |
| Max and Minnie Tomerlin Voelcker Fund | Young Investigator Award | David S Libich<br>Myron S Ignatius |
| University of Texas Health Science Center at San Antonio | T32CA148724 | Kunal Baxi |
| University of Texas Health Science Center at San Antonio | TL1TR002647 | Kunal Baxi |
| University of Texas Health Science Center at San Antonio | Greehey Graduate Fellowship in Children's Health | Nicole Rae Hensch<br>Paulomi Modi |
| Cancer Prevention and Research Institute of Texas | Training Award RP170345 | Nicole Rae Hensch<br>Amanda E Lipsitt |
| Hyundai Hope On Wheels | Young Investigator Grant | Amanda E Lipsitt |

The funders had no role in study design, data collection and interpretation, or the decision to submit the work for publication.

### Author contributions

Jiangfei Chen, Kunal Baxi, Data curation, Formal analysis, Validation, Investigation, Visualization, Methodology, Writing – review and editing; Amanda E Lipsitt, Data curation, Investigation, Writing – review and editing; Nicole Rae Hensch, Formal analysis, Validation, Visualization, Methodology, Writing – review and editing; Long Wang, Data curation, Formal analysis, Investigation, Writing – review and editing; Prethish Sreenivas, Conceptualization, Data curation, Software, Formal analysis, Validation, Investigation, Methodology; Paulomi Modi, Data curation, Software, Formal analysis, Validation, Investigation, Visualization, Writing – review and editing; Xiang Ru Zhao, Data curation, Formal analysis, Investigation, Methodology; Antoine Baudin, Data curation, Formal analysis, Visualization, Methodology, Writing – review and editing; Daniel G Robledo, Data curation, Writing – review and editing; Abhik Bandyopadhyay, Data curation, Methodology, Writing – review and editing; Aaron Sugalski, Gail E Tomlinson, Supervision, Writing – review and editing; Anil K Challa, Validation, Investigation, Visualization, Writing – review and editing; Dias Kurmashev, Data curation, Formal analysis, Methodology; Andrea R Gilbert, Resources, Data curation, Investigation, Writing – review and editing; Peter Houghton, Resources, Supervision, Writing – review and editing; Yidong Chen, Resources, Data curation, Software, Formal analysis, Investigation, Methodology, Writing – review and editing; Madeline N Hayes, Data curation, Formal analysis, Investigation, Methodology, Writing – review and editing; Eleanor Y Chen, Formal analysis, Validation, Writing – review and editing; David S Libich, Resources, Formal analysis, Validation, Visualization, Writing – review and editing; Myron S Ignatius, Conceptualization, Formal analysis, Supervision, Funding acquisition, Writing – original draft, Project administration, Writing – review and editing

## Author ORCIDs
Amanda E Lipsitt ⓘ http://orcid.org/0000-0002-3757-8493
Nicole Rae Hensch ⓘ http://orcid.org/0000-0001-9946-0995
Long Wang ⓘ http://orcid.org/0000-0002-7935-4148
David S Libich ⓘ http://orcid.org/0000-0001-6492-2803
Myron S Ignatius ⓘ http://orcid.org/0000-0001-6639-7707

## Ethics

Human subjects: Patient presenting with osteosarcoma signed a Consent to be part of a Repository, Epidemiology of Cancer in Children, Adolescents and Adults. In brief, this allowed for the storage of tissue, cataloging of medical information, and for research to be conducted from collected samples. The study's IRB number is HSC20080057H. Patient was informed of the risks and benefits. The umbrella study covering epidemiological study and patient-derived xenograft generation is IRB approved through UT Health San Antonio.

Animal studies were approved by the UT Health San Antonio Institutional Animal Care and Use Committee (IACUC) under protocol #20150015AR (mice) and #20170101AR (zebrafish). Zebrafish images were taken with specimens under tricaine anesthesia. Zebrafish tumor extraction was performed by administering high dose tricaine to minimize suffering.

## Decision letter and Author response
Decision letter https://doi.org/10.7554/eLife.68221.sa1
Author response https://doi.org/10.7554/eLife.68221.sa2

---

# Additional files

## Supplementary files
• Supplementary file 1. Table summarizing the main findings for wild-type and mutant *TP53* on tumor initiation, localization, proliferation, apoptosis, and ability to transactivate gene expression.

• Supplementary file 2. Table with RNAseq data for gene expression comparing three $kRAS^{G12D}$;$tp53^{-/-}TP53^{P153Δ}$;*GFP-positive* tumors to three $kRAS^{G12D}$; $tp53^{-/-}$;*GFP-positive* tumors.

• Supplementary file 3. Table with RNAseq data for gene expression comparing one head to two trunks ERMS expressing $kRAS^{G12D}$; $tp53^{-/-}$;*GFP*.

• Supplementary file 4. List of primers and guide RNAs used in this study.

• Supplementary file 5. List of antibodies used in this study.

• Supplementary file 6. List of reagents used in this study.

• Transparent reporting form

• Source data 1. Source data containing full immunoblots for the data presented in the figures.

## Data availability
Data sets were submitted to Dryad, available here: https://doi.org/10.5061/dryad.zgmsbccb6.

The following datasets were generated:

| Author(s) | Year | Dataset title | Dataset URL | Database and Identifier |
|---|---|---|---|---|
| Chen Y, Ignatius M | 2023 | TP53 Zebrafish Data | https://www.ncbi.nlm.nih.gov/geo/query/acc.cgi?acc=GSE213869 | NCBI Gene Expression Omnibus, GSE213869 |

*Continued on next page*

*Continued*

| Author(s) | Year | Dataset title | Dataset URL | Database and Identifier |
|---|---|---|---|---|
| Hensch NR, Chen J, Baxi K, Lipsitt AE, Wang L, Baudin A, Robledo DG, Bandyopadhyay A, Challa A, Gilbert AR, Tomlinson GE, Houghton P, Chen EY, Libich DS, Ignatius MS, Modi P, Zhao XR, Kurmashev D, Chen Y, Hayes MN, Sreenivas P | 2023 | Defining function of wild-type and three patient specific TP53 mutations in a zebrafish model of embryonal rhabdomyosarcoma | https://doi.org/10.5061/dryad.zgmsbccb6 | Dryad Digital Repository, 10.5061/dryad.zgmsbccb6 |

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
