## [Editor Report]

This paper uses the zebrafish as an in vivo model for exploring cancer genetics. The work on patient-specific alleles of the key oncogene TP53 enables new insights. The focus on embryonic stages enables a new understanding of the mechanism underlying this pediatric cancer.

---

## [Decision Letter]

**Decision letter after peer review:**

Thank you for submitting your article "Defining function of wild-type and patient specific TP53 mutations in a zebrafish model of embryonal rhabdomyosarcoma" for consideration by *eLife*. Your article has been reviewed by 3 peer reviewers, and the evaluation has been overseen by a Reviewing Editor and Richard White as the Senior Editor. The following individuals involved in review of your submission have agreed to reveal their identity: Maura McGrail (Reviewer #1).

Essential revisions:

1) Are the TP53 alleles really hypomorphic? The cell line work does not address this issue. It should be demonstrated that the proteins retain some level of transcriptional activity. qPCR of P53 target genes (BAX, BBC3/PUMA, PMAIP1/NOXA) in the zebrafish tumor model would address the mechanism by which the human variants can induce apoptosis.

2) The osteosarcoma cell line work should be removed as it is out of place in this manuscript, given the focus on embryonal rhabdomyosarcoma. These should be removed and if any in vitro studies are included, these should be in in an ERM cell line.

3) Language: For example, the authors overstate their claim in the Introduction, "highlight the zebrafish ERMS model as a powerful and high-throughput system" when they describe examining only 3 specific mutations in this report. This should be addressed throughout to reflect the focus of this work on embryonal rhabdomyosarcoma. Such focus will improve the communication of this story.

*Reviewer #1 (Recommendations for the authors):*

Results and Discussion:

1. The Langenau et al., 2007 rag2-kRASG12D ERMS model is well established and the results briefly describe the model, but it would be helpful to provide a more thorough description of the model in the first paragraph, particularly since this is a somatic model created by injecting linear DNA expressing rag2 promoter driven kRASG12D and a rag2 promoter driven fluorescent reporter. Data presented later in the manuscript underscores the considerable variability in generating tumors or driving gene expression using this approach. The concentration of DNA injected into single cell embryos to create tumors was indicated in the Methods, but the amount of DNA injected wasn't recorded. Please include. A diagram in figure 1 showing how the model is generated would also be very helpful.

2. Figure 1. Include the age of the individual fish shown in Figure 1. Some of the arrows point to tissue with DsRed expression, which may or may not be transformed by expression of kRASG12D. It would be useful to explain the criteria used to determine a tumor vs. simply expression of the co-injection marker.

3. Figure S1C-E The impact of loss of tp53 on the rate of ERMS tumor growth was assessed over a 3 week period using age matched fish. The age of the fish wasn't indicated, but they appear to be adults, not juveniles as shown in Figure 1. Overall, a more complete description of procedures and analyses would strengthen the paper.

4. Figure S1C-E The text states that metastases were observed in ERMS ; tp53-/- fish #3. A second site tumor isn't visible in the image of fish #3. To support this claim in the text, evidence for metastasis vs. occurrence of a second site primary tumor is required.

5. Figure 2 A and B. Cellular characterization of the ERMS; tp53-/- tumors reveals increased cellular proliferation compared to ERMS tumors. The numbers for panels A (Edu+ cells) and B (pHH3+ cells) appear to be switched: 2.3 vs. 13.3 should be in B, 21 vs 65.5 should be in A. Does n refer to individual tumors, or individual sections? The data is very strong, but quantification plots should be included with p values for both markers of proliferation.

6. Figure 2 C-E and Figure 3. The analysis of apoptosis in the ERMS vs. ERMS; tp53 tumors indicates a lack of apoptosis in the tumor model. Figure 3 presents data to show that expression of human Tp53 significantly suppressed tumor initiation and tumor size in ERMS; tp53-/- animals. The data is significant, and indicates a reduction in the number of EdU positive proliferating cells. Given ERMS tumors in a wildtype zebrafish tp53 background have very little apoptosis, the observation that there is an increase in apoptosis could be further clarified.

7. The comparison of ERMS;tp53-/- with and without expression of human TP53 showed that there isn't a significant difference in the number of tumors per fish. Data is needed to support the claim that expression of human tp53 prevents spreading and/or metastasis.

8. Figure S3: An experiment testing the dose dependency of tumor suppression was performed by injection of different concentrations of rag:TP53. 15ng/ul vs. 20ng/ul of linear construct was injected. Details on the volume injected, whether this was human tp53 or zebrafish tp53, were not included. In order to demonstrate a dose response, the data should show that there is a significant difference between 15 ng/ul vs. 20 ng/ul, and the experiment completed with at least 3 biological replicates. The data in c shows high variability in tp53 protein levels in individual tumors, without details on the source of the tumors – were they from fish injected with the same dose, or different doses. Overall this data suggests the method of the tumor model and expression of tp53 variants leads to significant inconsistencies. A more thorough analysis is necessary to draw a conclusion about dose dependency.

9. Figure 4 and Figure 5. The activity of tp53-C176F and tp53-P153∆ were analyzed in osteosarcoma cells, and it was observed neither inhibits cell growth, suppresses colony formation in vitro, drives luciferase expression, or induces apoptosis or expression of p21. This indicates neither mutant has wild type function, transcriptional activation activity, or is a gain of function.

When expressed in the zebrafish ERMS; tp53-/- model, tp53-C176F can suppress tumor formation (Figure 5B) and lead to increased apoptosis (Figure 5F) – a p value to demonstrate significance should be included. This doesn't match with the lack of apoptosis in ERMS; tp53+/+ data referenced in point 6. Including qPCR analysis of tp53 target gene expression in the zebrafish tumors would address whether tp53-C176F retains some level of wild type activity.

10. Section heading starting on page 9. Figure 5. The analyses of tp53-C176F and tp53-P153∆ could be placed in separate sections or described in separate paragraphs, so the results documenting effect on tumor incidence, location, effects on proliferation and apoptosis are clearly and systematically described for each mutant in comparison to wild type. In the text tp53-C176F is described as a hypomorph. tp53-P153∆ is described as a gain of function. But the figure legend for Figure 5 indicates both are gain of function.

Figure 5D. Does the incidence of head tumors occur in relatively young fish? The image in Figure 5D bottom panel appears to be a juvenile fish, in contrast to the adult fish shown in the top and middle panels. The scale bar isn't indicated in the figure legend. This could be an interesting result to follow up – a correlation between age of onset and tumor location.

11. Figure S5. The critical experiment to demonstrate tp53-C176F is a hypomorph that retains some wildtype function was designed as follows: tp53-C176F was expressed in the ERMS; tp53-/- model, then tumors from control and tp53-C176F were explanted into hosts and the hosts were treated with or without the tp53 stabilizer ZMC1. The representative images in panels D (ERMS; tp53-/- + tp53-C176F) and H (ERMS; tp53 -/-) aren't equivalent – both control groups treated with DMSO should show the same level of tumor growth/expansion; both ZMC1 treatment groups appear to impact tumor expansion equally.

12. In the Results section "Expression of TP53Y220C predisposes to head ERMS in zebrafish" starting on page 10 there is extensive discussion of the co-occurrence of tp53-P153∆ with tp53-Y220C. This would fit better in the introduction and/or in the discussion.

13. Figure 6 shows results examining the tp53-Y220C variant on tumor incidence, location, level of apoptosis and proliferation. The switch from co-expressing a DsRed marker to a GFP marker wasn't described.

Similar to the analysis of tp53-C176F and tp53-P153∆, the tp53-Y220C variant did not demonstrate wild type tp53 activity in in vitro assays, but had a suppressive effect on tumor incidence in the zebrafish ERMS;tp53-/- model. It also appeared to lead to increased incidence of head tumors. Figure 6 D and F doesn't indicate age or size of the fish shown.

14. Inclusion of a summary diagram or table that maps activities to each tp53 mutant/variant analyzed would help to clarify the key findings of the study.

15. Overall, there was an absence of detail in the figure legends, number of biological replicates for key experiments, age of fish shown in the images, and scale bar measurements.

16. Discussion:

The discussion could better address why overexpression of tp53 variants in the zebrafish ERMS; tp53-/- model could suppress tumor initiation, with or without impact on proliferation and apoptosis, when the same tp53 variants did not have an impact on osteosarcoma cell proliferation, viability, and colony formation in vitro. The conclusion that the variants are either hypomorphs or gain of function alleles would be better supported by demonstrating their effect on gene expression or other cellular activity.

A number of statements were made that are vague or not supported by the data:

"when compared to wild-type animals, tp53-/- animals … exhibit a relatively less pronounced effect on tumor cell apoptosis." "Thus, while TP53 is shown to suppress tumor progression by multiple mechanisms depending on tumor type, our results suggest that loss of TP53 in human ERMS may increase aggressiveness through enhanced proliferation, invasion, and metastasis." Evidence of invasion or metastasis in the tumor model aren't presented.

*Reviewer #2 (Recommendations for the authors):*

1. The authors overstate their claim in the Introduction, "highlight the zebrafish ERMS model as a powerful and high-throughput system" when they describe examining only 3 specific mutations in this report.

2. More details regarding the transgenic generation approach should be described and the transgenic constructs clearly articulated throughout the Results section. The authors shouldn't assume that readers are familiar with prior strategies undertaken by these authors or by the Langenau lab.

3. Page 6: in the phrase "Treatment of wild-type and tp53-/- tumors with a 6-hour pulse of EdU" – it needs to be clarified that this is being undertaken in whole larval context.

4. The concepts of early and late apoptosis need to be properly defined and explained.

5. Page 6: the phrase "introducing wild-type human TP53 (TP53WT) in ERMS tumors in tp53-/- animals" is not an accurate reflection of what is being undertaken, which is introducing WT TP53 along with the KRAS transgene at the outset rather than adding the gene to already developed tumors, which is how the text currently reads.

6. The authors should comment on the amino acid conservation between human and zebrafish p53 and how this may impact interactions with other endogenous zebrafish proteins.

7. Page 8: the statement that the TP53 mutants tested "do not show any gain-of function activity in the SaOS2 TP53-/- background" appears a generalization not justified by the experimental data presented.

8. The Discussion should reference that the order in which oncogenic mutations occur may impact tumor phenotype and as such, may not be fully reflected in the zebrafish models as presented.

*Reviewer #3 (Recommendations for the authors):*

1. In many figures, including Figures 1D, 3C, S1E, and S5G, the plots of Tumor/body size or Tumor area ratio showed very high tumor burden in certain groups, the ratios in some plots were even higher than 0.6. Please make sure your studies complied with the IACUC regulation on the tumor size/weight per animal.

2. Although the SaOS2 cell line is a commonly used assay system to study effects of TP53 variants, the data generated in this cell line were not consistent at all with those obtained from the in vivo zebrafish studies. Could authors discuss this inconsistency? Can authors use ERM cell line(s) instead of the SaOS2 osteosarcoma cell line to assess the activities of the TP53P153△, TP53C176F and TP53Y220C variants? In addition, engineering the cell lines with these exact mutations, not overexpression, would be helpful to accurately characterize their activities in vitro.

3. Coinjection of either TP53P153△ or TP53Y220C mutant with kRASG12D can increase development of ERMS in the head musculature. However, these two mutants behave very differently in many aspects, such as in kRASG12D-induced tumor initiation (Figures 5B vs. 6C) or in cell survival of ERMs (Figures 5F vs. 6I). Can authors provide clearer explanation on how these two different mutants promote more ERM tumorigenesis in the head musculature? Since it has been hypothesized that both TP53P153△ and TP53Y220C mutants might cause p53 instability, it would be interesting to test whether ZMC1 can stabilize these mutants leading to inhibition of head tumor development. In addition, whether Hedgehog signaling pathway could be specifically activated in the head tumors with overexpression of kRASG12D and TP53P153△ or TP53Y220C?

4. TP53C176F coinjection with kRASG12D in p53-/- fish can reduce kRASG12D-induced tumorigenesis (Figure 5B) and increase apoptosis (Figure 5F), suggesting TP53C176F functions as a hypomorphic mutant. Can authors examine whether the p21 expression is upregulated in the ERMs with TP53C176F overexpression, which would provide further mechanistic insight on the hypomorphic effect of the TP53C176F mutant.

[Editors' note: further revisions were suggested prior to acceptance, as described below.]

Thank you for resubmitting your work entitled "Defining function of wild-type and three patient specific TP53mutationsin a zebrafish model of embryonal rhabdomyosarcoma" for further consideration by *eLife*. Your revised article has been evaluated by Richard White (Senior Editor) and a Reviewing Editor.

There is strong consensus that the manuscript has been substantially improved but there are some remaining issues that need to be addressed, as outlined below:

First, there appears to be a missing item – Supplemental Table I. Second, the new findings about the occurrence of medulloblastoma in the tp53P153 model should be mentioned in the Abstract.

*Reviewer #1 (Recommendations for the authors):*

Overall, the revised manuscript addressed the essential revisions and the authors have responded to the reviewer's requests. The revised manuscript also presents two new analyses, which weren't mentioned in the abstract or introduction. A new Figure 5 and supporting figures show identification of medulloblastoma in addition to head ERMS arising in the tp53P153delta ERMS model. A new Figure 7 shows experiments to test the dependence of the tp53P153delta ERMS tumor induction on Kdr, which is overexpressed in patient osteosarcoma and the zebrafish model. The Kdr gene and function of the protein could be better introduced, and the experiment have stronger experimental validation as described below. But the kdr experiment illustrates how this model can be used to investigate the in vivo mechanisms by which patient tp53 variants drive ERMS tumorigenesis.

Some issues were not adequately addressed: the statement that the model is high throughput has not been removed from the discussion, Figure 2 A, B quantification data was not included and it's still unclear what the numbers in those panels reflect, and new RT-qPCR on p53 target gene expression contained contradictory or inconsistent results. Different tp53 variants did not lead to consistent elevation of apoptotic genes bax, bbc3, noxa, – tp53C176F and tp53P153delta induced expression of bbc3, but not noxa or bax; tp53Y220C increased noxa and cell growth arrest gene gadd54a; cell cycle arrest gene cdkn1a was elevated in all variant models including the tp53 null. A discussion of this data, in the context of demonstrating that the tp53 variants retain some level of transcriptional activity, was not included.

The experiment to demonstrate ERMS tp53P153delta is dependent on elevated kdr expression via somatic CRISPR RNP kdr targeting did not contain much detail. A more thorough analysis of kdr deletion/mutagenesis in the tumors, kdr transcript knockdown measured by RT-qPCR, would validate the conclusion that Kdr drives oncogenesis in the ERMS tp53P153delta model. Figure 7 Supplement 1 does not contain enough detail explaining how targeted mutagenesis was induced, documented and measured.

*Reviewer #2 (Recommendations for the authors):*

The authors have done an admirable job responding to previous reviews, resulting in a manuscript that is more logically constructed and better focused in the data presented.

A few residual comments:

1. I was not able to find the Supplemental Table that was suggested by reviewers and referred to in the Response to Reviewers that summarizes and highlights the main findings and differences between wild type and various tp53 zebrafish mutants described in the manuscript.

2. The choice of analyzing the TP53 P153 mutation remains a bit unusual given its rarity and identification as a germline mutation in an aggressive osteosarcoma, given the focus of this manuscript on ERMS (especially now with removal of the osteosarcoma cell line data which contrasted with the zebrafish data).

3. Moreover this P153 mutation leads to Shh medulloblastoma tumors in a subset of zebrafish. This is new data that is interesting, but a bit of a distraction from the focus of the manuscript on ERMS.

---

## [Author Response]

Essential revisions:1) Are the TP53 alleles really hypomorphic? The cell line work does not address this issue. It should be demonstrated that the proteins retain some level of transcriptional activity. qPCR of P53 target genes (BAX, BBC3/PUMA, PMAIP1/NOXA) in the zebrafish tumor model would address the mechanism by which the human variants can induce apoptosis.

As requested, qPCR was performed on newly generated primary tumors expressing *tp53^+/+^, tp53^-/-^* and *tp53^-/-^* + *TP53^C176F^*, *tp53^-/-^* + *TP53^P153^*^Δ^ and *tp53^-/-^* + *TP53^Y220C^* and p53 target genes including *baxa, cdkn1a*, *gadd45a*, *noxa* and *puma/bbc3* were assessed. We report that in the *TP53^C176F^* and *TP53^P153^*^Δ^ expressing tumors *bbc3* is induced but not *noxa, baxa* or *gadd45a.* While in the *TP53^Y220C^* expressing tumors *gadd45a* is induced and in 2 of the 3 primary tumors *noxa* is induced. Rather unexpectedly in the *tp53^-/-^, tp53^-/-^* + *TP53^C176F^*, *tp53^-/-^* + *TP53^P153^*^Δ^ and *tp53^-/-^* + *TP53^Y220C^* tumors *cdkn1a* is induced compared to *tp53^+/+^* tumors. We summarize our major findings in Supplemental table 1. Similarly, we find in human RMS cell lines high p21/CDKN1A is expressed in multiple cell lines which are both wild-type or mutant for p53 (see Reviewer figure 3). Suggesting that in RMS CDKN1A expression can also be independent of p53.

We also generated new tumors to address reviewer comments. Unexpected, we found that the *tp53^-/-^* + *TP53^P153^*^Δ^ head tumors also included potential brain tumors which resembled medulloblastomas (new Figure 5). These tumors have previously never been reported in the *kRAS^G12D^* ERMS model in zebrafish but can occur in Li Fraumeni patients. As this finding could confound our reported results, we assessed HandE of tumors in our analyses and also generated new tumors to assess the contribution of the medulloblastoma to our analyses and show that a majority of (16 of 21) of head tumors are ERMS. Lastly, we characterize the head ERMS tumors and we show that (i) Head ERMS tumors are molecularly distinct from trunk tumors and (ii) *kdr* downstream of *TP53^P153^*^Δ^ predisposes to head ERMS in tp53^-/-^ zebrafish.

2) The osteosarcoma cell line work should be removed as it is out of place in this manuscript, given the focus on embryonal rhabdomyosarcoma. These should be removed and if any in vitro studies are included, these should be in in an ERM cell line.

We agree with the reviewers and have removed in vitro studies in the SaOS2 cells.

3) Language: For example, the authors overstate their claim in the Introduction, "highlight the zebrafish ERMS model as a powerful and high-throughput system" when they describe examining only 3 specific mutations in this report. This should be addressed throughout to reflect the focus of this work on embryonal rhabdomyosarcoma. Such focus will improve the communication of this story.

We have made changes throughout the manuscript to reflect only what our results find and have made changes recommended to the text and Results section (Please see reviewer comments that address specific statements).

Reviewer #1 (Recommendations for the authors):Results and Discussion:1. The Langenau et al., 2007 rag2-kRASG12D ERMS model is well established and the results briefly describe the model, but it would be helpful to provide a more thorough description of the model in the first paragraph, particularly since this is a somatic model created by injecting linear DNA expressing rag2 promoter driven kRASG12D and a rag2 promoter driven fluorescent reporter. Data presented later in the manuscript underscores the considerable variability in generating tumors or driving gene expression using this approach. The concentration of DNA injected into single cell embryos to create tumors was indicated in the Methods, but the amount of DNA injected wasn't recorded. Please include. A diagram in figure 1 showing how the model is generated would also be very helpful.

We thank the reviewer for this suggestion and agree that schematics setting up the experiments described in the zebrafish ERMS model are important. We have therefore included schematics for the main figures to better explain the experimental set up. We also have included in the Methods section the amount of DNA injected in addition to the concentration of the DNA cocktails used in different experiments.

Amount of DNA in experiments. Wild-type AB, tp53^-/-^ AB and tp53^-/-^ AB/CG1 mixed strain zebrafish can tolerate a maximum of approximately 25-30 pg of DNA, while CG1, tp53-/-; CG1 strain syngeneic zebrafish can tolerate half that amount (15 pg of DNA). Second, the amount of kRAS^G12D^ or *TP53* wild-type or mutant total DNA injected is kept constant with a mix of 40 ng *rag2-kRAS^G12D^*/ 20 ng *rag2-dsRED/GFP (*60 ng/μL*)* in experiments in Figure 1 and for the rest of the manuscript 35 ng *rag2-kRAS^G12D^*/15 ng *rag2-TP53* wild-type or mutant/ 10 ng *rag2-GFP/dsRED* (60 ng/μL).

Addressing variability in generating tumors. As ERMS tumors arise very quickly within 60 days (2 months) and animals can start breeding on average between 4-6 months, it is not possible to maintain stable rag2:kRAS^G12D^ stable transgenic lines. We have attempted to generate conditional transgenic lines (UAS-Gal4 transgenics) in the past and while we have shown that the method works in principle. However, we were unable isolate stable lines that generate tumors possibly due to silencing of the kRAS^G12D^ in the germline. To account for variability in our experiments in each figure we perform both experimental and control arms of the experiment at the same time. i.e. on any given day we inject zebrafish (100 to >500) embryos from zebrafish mattings for both experimental groups and raise animals in the exact same conditions. We also perform at least 2 or more independent micro-injection experiments for each analyses. As there are loses of larval zebrafish due to normal animal husbandry, we combine data from experiments that were done close together i.e. within the same 4-8 week period. Thus, while the ERMS in zebrafish model is relatively easy use, each Kaplan Meijer analyses in the figure panel takes on average 3-6 months to complete and for each Kaplan Meijer analyses we have > 100 larval zebrafish per group except for Figure 7F (n>45 in both experimental arms). Additionally, because tumor initiation in the tp53^-/-^ background results in >75-97% of the experimental zebrafish initiating tumors it is possible to study more subtle differences between human *TP53* mutants. However, in the zebrafish tp53 wild-type setting, where tumor initiation can vary to about 15-35% of zebrafish getting tumors, it is extremely difficult to discern subtle differences in ERMS initiation. We would like to state that our protocol is a similar to protocols used by the laboratories of Dr. David Langenau, Dr. Thomas Look, Dr. Leonard Zon and former members from these laboratories to generate ERMS, leukemia, neuroblastoma, melanoma and other tumors. Lastly, in murine or zebrafish in vivo tumor models where tumors can take 10 days to a year or more to initiate, biological replicates are not the norm but rather a power calculation used to assess number of animals needed to achieve a p value of <0.05 compared to experimental and the control group.

2. Figure 1. Include the age of the individual fish shown in Figure 1. Some of the arrows point to tissue with DsRed expression, which may or may not be transformed by expression of kRASG12D. It would be useful to explain the criteria used to determine a tumor vs. simply expression of the co-injection marker.

We have updated all our figures and provide age of animals. We have previously shown that rag2:dsRED is highly expressed in all ERMS tumors, including in tumors in zebrafish that are < 10 days old. In the muscle, *rag2:dsRED* is expressed at very low levels which is not easily observable in non-tumor muscle cells even in the *Tg*(*rag2-GFP)* or *Tg (rag2-dsRED)* transgenic animals and hence easy to exclude from our analyses. We have previously confirmed tumors and developed methods to stage our ERMS tumors in experiments and show by co-labeling that even the earliest rag2:DsRED tumors that occur in one half or single somites express high levels of tumor propagating cell marker *myf5-GFP* and these early stage tumors were also confirmed via histology (Ignatius et al., Cancer Cell 2012). In our manuscript, we are being conservative and calling only stage 2 and above tumors, we also confirm tumors by performing HandE staining on a subset of all our tumors in all figures/experiments and the histology has been confirmed by soft tissue expert pathologist Dr. Eleanor Chen.

3. Figure S1C-E The impact of loss of tp53 on the rate of ERMS tumor growth was assessed over a 3 week period using age matched fish. The age of the fish wasn't indicated, but they appear to be adults, not juveniles as shown in Figure 1. Overall, a more complete description of procedures and analyses would strengthen the paper.

Please see below.

4. Figure S1C-E The text states that metastases were observed in ERMS ; tp53-/- fish #3. A second site tumor isn't visible in the image of fish #3. To support this claim in the text, evidence for metastasis vs. occurrence of a second site primary tumor is required.

We agree with the reviewer, the experiments we have done in the primary tumor setting cannot exclude the possibility that tumors occurring at secondary sites are not a primary tumor at the secondary site. Since suppression of invasion and metastasis effects of wild-type tp53 have already been previously established in a transplant model using syngeneic zebrafish (Ignatius et al., *eLife* 2018) and invasion and metastasis is not directly relevant to our current study, we have removed this analyses from our revised manuscript. Secondly, as tumors in tp53^-/-^ mutants initiate in younger animals and grow rapidly we had to pick from the subset that occur later and are smaller slower growing in order to carry out this analyses.

5. Figure 2 A and B. Cellular characterization of the ERMS; tp53-/- tumors reveals increased cellular proliferation compared to ERMS tumors. The numbers for panels A (Edu+ cells) and B (pHH3+ cells) appear to be switched: 2.3 vs. 13.3 should be in B, 21 vs 65.5 should be in A. Does n refer to individual tumors, or individual sections? The data is very strong, but quantification plots should be included with p values for both markers of proliferation.

We thank the reviewer in pointing out this discrepancy. We have updated our figures and figure legends to point out how these counts were performed. In panels A and B, we are estimating the percentage of Edu+ cells compared to the total number of DAPI-positive nuclei which are also dsRED-positive (Tumor). In B we calculate the total number of pHH3-positive nuclei in 6 confocal sections. Since much fewer cells are actively undergoing mitosis compared to those in S phase (Edu labeling), this sampling is able to overcome the variability of performing counts on single sections. In the figure we show one section.

6. Figure 2 C-E and Figure 3. The analysis of apoptosis in the ERMS vs. ERMS; tp53 tumors indicates a lack of apoptosis in the tumor model. Figure 3 presents data to show that expression of human Tp53 significantly suppressed tumor initiation and tumor size in ERMS; tp53-/- animals. The data is significant, and indicates a reduction in the number of EdU positive proliferating cells. Given ERMS tumors in a wildtype zebrafish tp53 background have very little apoptosis, the observation that there is an increase in apoptosis could be further clarified.

We thank the reviewer for pointing this out. We have included a statement in the text to clarify this difference. We believe the increased apoptosis seen in tumors co-expressing zebrafish or human WT *TP53* is due to higher levels of the wild-type protein. This is a limitation of our experimental set up, but it is also the approach that is commonly used to study effects of oncogenes, for example, Myc, Notch, Akt2, or tumor suppressors for example, *bbc3/ puma*, *noxa* on tumor biology in zebrafish cancer models (Blackburn et al., 2014; Langenau et al., 2007; Lobbardi et al., 2017; White, Rose, and Zon, 2013). In order to ensure we are not expressing high levels of wild-type and mutant p53 proteins, we compared the amount of mutant or wild-type protein expressed to mutant p53 in the Rh30 rhabdomyosarcoma cell line. Rh30 cells express the p53^R273C^ hotspot mutant protein. It is important to note that mutant p53 proteins can be highly expressed in tumors and the p53^C176F^ and p53^P153Δ^ mutant proteins studied here are highly expressed via IHC and western blot analyses (see Figure 4C, D). Further, our data also show that methods that can reactivate wild-type or mutant protein can also effect apoptosis if induced sufficiently.

7. The comparison of ERMS;tp53-/- with and without expression of human TP53 showed that there isn't a significant difference in the number of tumors per fish. Data is needed to support the claim that expression of human tp53 prevents spreading and/or metastasis.

Since in primary tumors we are unable to study metastasis and the data we have is mostly corelative, we have removed invasion and metastasis from the manuscript and focus on effects on tumor initiation, tumor number, growth, proliferation, apoptosis and site of tumor initiation. We do find that compared to tumors that initiate in the tp53^-/-^ background, overall fewer *TP53* or *Tp53* expressing animals initiate tumors. However, in the tp53^-/-^ background in zebrafish expressing rag2:kRAS^G12D,^ rag2:TP53 and rag2:dsRED in tumor burdened zebrafish multiple tumors can initiate per animal suggesting possibly non-cell autonomous roles for *tp53* in tumor initiation.

8. Figure S3: An experiment testing the dose dependency of tumor suppression was performed by injection of different concentrations of rag:TP53. 15ng/ul vs. 20ng/ul of linear construct was injected. Details on the volume injected, whether this was human tp53 or zebrafish tp53, were not included. In order to demonstrate a dose response, the data should show that there is a significant difference between 15 ng/ul vs. 20 ng/ul, and the experiment completed with at least 3 biological replicates. The data in c shows high variability in tp53 protein levels in individual tumors, without details on the source of the tumors – were they from fish injected with the same dose, or different doses. Overall this data suggests the method of the tumor model and expression of tp53 variants leads to significant inconsistencies. A more thorough analysis is necessary to draw a conclusion about dose dependency.

We agree with the reviewer and have removed the 20 ng/μL arm from our analyses and include data only for the 15 ng/μL arm as this corresponds to the amount of *TP53* wild-type and mutant DNA we use in all our analyses. All western blots analyses were from the 15 ng/μL injections of the small tumors that arise from this injection. The zebrafish Tp53 in #2 tumor is higher than others, however this in outlier in expression of p53 compared to all our experiments. please see figure 3A, 4G and 6B.

9. Figure 4 and Figure 5. The activity of tp53-C176F and tp53-P153∆ were analyzed in osteosarcoma cells, and it was observed neither inhibits cell growth, suppresses colony formation in vitro, drives luciferase expression, or induces apoptosis or expression of p21. This indicates neither mutant has wild type function, transcriptional activation activity, or is a gain of function.

We agree with the reviewers that the studies in the SaOS2 cells which are *TP53* deficient osteosarcoma cells does not agree in vivo studies in our zebrafish embryonal rhabdomyosarcoma model. We would like to note that while the SaOS2 and other *TP53* deficient cells have traditionally used to characterize mutant *TP53* function a significant subset of mutants do not show any phenotypes or show mutant p53 effects only in some and not in other assay systems. Additionally, whether *TP53* mutants can have GOF effects remains controversial in some tumors for example leukemia (Boettcher et al., 2019). We initially added the SaOS2 data to highlight that the SaOS2 cell based system may not be optimal for the mutants we study. However, as we have not tested other cell systems and have not generated knock-in alleles for these mutants in ERMS cells lines, we cannot fully support this assertion and have accordingly removed the SaOS2 data as suggested.

When expressed in the zebrafish ERMS; tp53-/- model, tp53-C176F can suppress tumor formation (Figure 5B) and lead to increased apoptosis (Figure 5F) – a p value to demonstrate significance should be included. This doesn't match with the lack of apoptosis in ERMS; tp53+/+ data referenced in point 6. Including qPCR analysis of tp53 target gene expression in the zebrafish tumors would address whether tp53-C176F retains some level of wild type activity.

We have included this data in Figure 5F (now Figure 4K) and show that tumors expressing *TP53^C176F^* in the tp53^-/-^ background compared to tp53^-/-^ tumors. However, there is no difference in the rate of apoptosis between *tp53^-/-^* and *TP53^P153^*^Δ^; *tp53^-/-^* expressing tumors. Mutant p53 proteins can often be highly expressed in tumors compared to the WT p53 which is usually poorly expressed but can be induced after damage. Additionally, we do show that the WT human p53 can complement the zebrafish protein and that the WT and the three mutants can show very different but reproducible effects on different aspects of *TP53* function.

As requested qPCR was performed on newly generated primary tumors. “To assess the effects of p53 variant expression on transcription, we performed qPCR analyses comparing *tp53^+/+^, tp53^-/-^* and *tp53^-/-^* + *TP53^C176F^* and *tp53^-/-^* + *TP53^P153^*^Δ^ ERMS and p53 target genes including *baxa, cdkn1a*, *gadd45a*, *noxa* and *puma/bbc3*. We found that both *tp53^-/-^*+*TP53^C176F^* and *tp53^-/-^* + *TP53^P153^*^Δ^ tumors displayed increased *bbc3* expression compared to *tp53^-/-^* or *tp53^+/+^* tumors, but had no difference with respect to *cdkn1a*, *gadd45a* and *noxa* expression. We also found that *tp53^-/-^* + *TP53^C176F^* and *tp53^-/-^* + *TP53^P153^*^Δ^ tumors expressed higher levels of *cdkn1a* compared to *tp53^+/+^* ERMS; however, rather unexpectedly *tp53^-/-^* tumors also express higher levels of *cdkn1a* compared to *tp53^+/+^* controls (Figure 4 Supplement 3).”

10. Section heading starting on page 9. Figure 5. The analyses of tp53-C176F and tp53-P153∆ could be placed in separate sections or described in separate paragraphs, so the results documenting effect on tumor incidence, location, effects on proliferation and apoptosis are clearly and systematically described for each mutant in comparison to wild type.

We have rewritten this section and have separated out effects of the *TP53^C176F^and TP53^P153^*^Δ^ mutants in the text. Additionally, we provide a new table (Supplemental Table 1) to compare and contrast the cellular effects of the wild-type and mutant *TP53* genes when expressed in the zebrafish ERMS model.

In the text tp53-C176F is described as a hypomorph. tp53-P153∆ is described as a gain of function. But the figure legend for Figure 5 indicates both are gain of function.

We thank the reviewer for pointing out this oversight and have changed the figure legend to “*TP53^C176F^* is a hypomorphic allele while *TP53^P153^*^Δ^ has gain-of-function effects in ERMS”.

Figure 5D. Does the incidence of head tumors occur in relatively young fish? The image in Figure 5D bottom panel appears to be a juvenile fish, in contrast to the adult fish shown in the top and middle panels.

We went back to our analyses and find that the head tumors do not initiate later. However, we do find that the head tumors grow slightly more slowly. To clearly reveal the head tumor phenotypes, we waited until the tumors were more prominent. This is consistent with our finding that the tumors in the tp53^-/-^ background come up quickly and grow rapidly. We also generated head ERMS tumors expressing *TP53^P153^*^Δ^ in the syngeneic CG1; tp53^-/-^ background. We expanded 3 head tumors expressing *TP53^P153^*^Δ^ and also 3 tp53^-/-^ tumors (1 head and 2 trunk tumors), sorted pure secondary tumors and performed RNAseq analyses across both groups. Rather unexpectedly, the head tumors we isolated were all pro-neural brain tumors with gene signatures consistent with the sonic hedgehog sub-group of medulloblastoma (See new Figure 5). To address how frequent the medulloblastoma tumors were, we generated additional tumors with the following genotypes kRAS^G12D^/ tp53^-/-^, kRAS^G12D^/ *TP53^P153^*^Δ^, tp53^-/-^ and kRAS^G12D^/ TP53^C176F^, tp53^-/-^. We identified 5 out 21 head tumors had histology consistent with medulloblastoma in the kRASG12D/ *TP53^P153^*^Δ^, tp53^-/-^ group, the rest were all head ERMS tumors (>76% of all head tumors). In contrast only 1 out of 20+ kRAS^G12D^/ tp53^-/-^ tumors had a medulloblastoma tumor and we did not see any head tumors in the kRAS^G12D^/ *TP53^C176F^*/tp53-/- group (0 of 20 ERMS). Finally, we compared the CG1; tp53^-/-^ ERMS head tumor with the two trunk tumors and show that the head tumors express genes consistent with head musculature including *tbx1, dlx3b, dlx4b* and *ptch2*, suggesting that there are molecular differences between head and trunk ERMS tumors consistent with analyses in a hedgehog driven mouse tumor model of head and neck rhabdomyosarcoma (Drummond et al., 2018).

The scale bar isn't indicated in the figure legend. This could be an interesting result to follow up – a correlation between age of onset and tumor location.

We have added scale bars and the age of the zebrafish in every figure or in the figure legend.

11. Figure S5. The critical experiment to demonstrate tp53-C176F is a hypomorph that retains some wildtype function was designed as follows: tp53-C176F was expressed in the ERMS; tp53-/- model, then tumors from control and tp53-C176F were explanted into hosts and the hosts were treated with or without the tp53 stabilizer ZMC1. The representative images in panels D (ERMS; tp53-/- + tp53-C176F) and H (ERMS; tp53 -/-) aren't equivalent – both control groups treated with DMSO should show the same level of tumor growth/expansion; both ZMC1 treatment groups appear to impact tumor expansion equally.

We have found that transplanted secondary tumors can grow at different rates. Taking this into account the variability in this experiment, we compare the effect of ZMC1 vs DMSO in *tp53^-/-^; TP53^C176F^* and *tp53^-/-^* tumors. Relative to the DMSO treatment there the ZMC1 *tp53^-/-^; TP53^C176F^* are smaller, have p53 stabilized and also have increased Annexin V-positive cells. In contrast, there is no difference in tumor size and apoptosis between DMSO and ZMC1 treated *tp53^-/-^* tumors.

12. In the Results section "Expression of TP53Y220C predisposes to head ERMS in zebrafish" starting on page 10 there is extensive discussion of the co-occurrence of tp53-P153∆ with tp53-Y220C. This would fit better in the introduction and/or in the discussion.

We have edited the text to briefly introduce why we chose to study the *TP53^Y220C^* mutant in zebrafish. We have moved the more detailed description to the Discussion section as suggested.

13. Figure 6 shows results examining the tp53-Y220C variant on tumor incidence, location, level of apoptosis and proliferation. The switch from co-expressing a DsRed marker to a GFP marker wasn't described.

We include a statement in the main text and also in the methods section addressing this change. We and other have interchangeably use dsRed and or GFP to label ERMS tumors (Ignatius et al., 2012; Ignatius et al., 2018; Langenau et al., 2008).

Similar to the analysis of tp53-C176F and tp53-P153∆, the tp53-Y220C variant did not demonstrate wild type tp53 activity in in vitro assays, but had a suppressive effect on tumor incidence in the zebrafish ERMS;tp53-/- model. It also appeared to lead to increased incidence of head tumors. Figure 6 D and F doesn't indicate age or size of the fish shown.

Please see Section 9.

We have added age of the zebrafish in Figure 6 D and F.

14. Inclusion of a summary diagram or table that maps activities to each tp53 mutant/variant analyzed would help to clarify the key findings of the study.

We thank the reviewer for this suggestion and have included a supplementary table (Supplemental Table 1) highlighting the main findings and differences between the wild-type and mutant *TP53* highlighted.

15. Overall, there was an absence of detail in the figure legends, number of biological replicates for key experiments, age of fish shown in the images, and scale bar measurements.

We thank the reviewer for this comment and have gone through each section of the manuscript and have included additional details including schematics for the experimental set up, included more details of how the experiments were performed, age of the fish in experiments, scale bar measurements and details of biological replicates. We have also highlighted these details in the response to reviewer points.

16. Discussion:The discussion could better address why overexpression of tp53 variants in the zebrafish ERMS; tp53-/- model could suppress tumor initiation, with or without impact on proliferation and apoptosis, when the same tp53 variants did not have an impact on osteosarcoma cell proliferation, viability, and colony formation in vitro. The conclusion that the variants are either hypomorphs or gain of function alleles would be better supported by demonstrating their effect on gene expression or other cellular activity.A number of statements were made that are vague or not supported by the data:"when compared to wild-type animals, tp53-/- animals … exhibit a relatively less pronounced effect on tumor cell apoptosis." "Thus, while TP53 is shown to suppress tumor progression by multiple mechanisms depending on tumor type, our results suggest that loss of TP53 in human ERMS may increase aggressiveness through enhanced proliferation, invasion, and metastasis." Evidence of invasion or metastasis in the tumor model aren't presented.

We have reworked/reworded the Results section and the discussion to include the following as recommended by the reviewers. (1) We have removed all data in SaOS2 p53 deficient cells. (2) We have removed data pertaining to invasion and metastasis as this data is corelative and not related to our main findings. We instead cite our previous publication where we addressed the role of *tp53* in suppressing invasion and metastasis and having no effect on tumor self-renewal (Ignatius et al., 2018). (3) We add a section about the limitations of our study.

Reviewer #2 (Recommendations for the authors):1. The authors overstate their claim in the Introduction, "highlight the zebrafish ERMS model as a powerful and high-throughput system" when they describe examining only 3 specific mutations in this report.

We thank the reviewer from their comment and agree. We have changed the title and the text to reflect the results presented to remove any statement that overrepresents our significant findings.

Title: Defining function of wild-type and three patient specific *TP53* mutations in a zebrafish model of embryonal rhabdomyosarcoma.

2. More details regarding the transgenic generation approach should be described and the transgenic constructs clearly articulated throughout the Results section. The authors shouldn't assume that readers are familiar with prior strategies undertaken by these authors or by the Langenau lab.

We thank the reviewers for this suggestion and agree that schematics setting up the experiments described in the zebrafish ERMS model is important. We have therefore included schematics for the main figures to better explain the experimental set up. We also have included in the Methods section the amount of DNA injected in addition to the concentration of the DNA cocktails used in different experiments.

3. Page 6: in the phrase "Treatment of wild-type and tp53-/- tumors with a 6-hour pulse of EdU" – it needs to be clarified that this is being undertaken in whole larval context.

We thank the reviewer for this suggestion and have clarified the experiment being performed.

“Wild-type and *tp53^-/-^* tumor burdened animals were treated with a 6-hour pulse of EdU and then animals were euthanized and sectioned, and stained for Edu-positive cells, a marker for proliferation.”

4. The concepts of early and late apoptosis need to be properly defined and explained.

We have added more details and a reference to introduce how Annexin V in combination with Propidium iodide can be used to label live cells that are progressing through apoptosis.

“Annexin V staining that can distinguish cells beginning to undergo apoptosis (early apoptosis; low Annexin V/Propidium Iodide (PI)-positive), or that are either undergoing apoptosis (late apoptosis; high Annexin V/high PI) or necrosis (high Annexin V/ low PI), was performed on live single cells suspensions of ERMS tumor cells extracted post euthanasia.”

5. Page 6: the phrase "introducing wild-type human TP53 (TP53WT) in ERMS tumors in tp53-/- animals" is not an accurate reflection of what is being undertaken, which is introducing WT TP53 along with the KRAS transgene at the outset rather than adding the gene to already developed tumors, which is how the text currently reads.

We agree with the reviewer comment and have amended the text to reflect what the experiment is testing.

“We next assessed the consequence of co-expressing wild-type human *TP53* (*TP53^WT^*) in the cells from which ERMS tumors initiate in *tp53^-/-^* animals. Co-expression of *kRAS^G12D^* along with *TP53^WT^* results in the expression of WT *TP53* from the very beginning in cells from which tumors initiate and also in the resulting tumors.”

6. The authors should comment on the amino acid conservation between human and zebrafish p53 and how this may impact interactions with other endogenous zebrafish proteins.

We have included the following statement and additional Supplemental figure in the Results section to address this comment by the reviewer.

“Zebrafish and human p53 are functionally similar and share 56% identity with respect to amino acid sequence (67% positives, Figure 3 Supplement 1) (Berghmans et al., 2005; Ignatius et al., 2018; Parant, George, Holden, and Yost, 2010; Storer and Zon, 2010). Within the core DNA-binding region where a majority of mutations occur in patients, 72% conservation exists (79% positives, Figure 3 Supplement 1, 2). To assess functional conservation in vivo, we co-expressed wild-type human *TP53* (*TP53^WT^*) in the cells from which ERMS tumors initiate in *tp53^-/-^* animals (Figure 3 Supplement 3). Importantly, co-expression of *kRAS^G12D^* along with *TP53^WT^* resulted in the expression of wild type *TP53* from the very beginning in cells from which tumors initiate and also in the resulting tumors.”

Secondly, in the Discussion section we have added a statement on the limitation of our in vivo model.

“Finally, we would like to state the limitation of our zebrafish ERMS model. In Li Fraumeni patients, *TP53* mutations are germline with somatic modifier mutations occurring secondarily. In our experiments, while our starting point is a tp53^-/-^ null background, *kRAS^G12D^* and the human *TP53* variant allele are introduced at the same time, under the control of the *rag2* promoter. A second limitation is that effects of mutant *TP53* in the tumor microenvironment are missed in our assays. However, our assay is very rapid, given that the generation of zebrafish strains with germline knock-in of the equivalent mutations takes an average of two years for full assessment. In mice, only the most common *Tp53* variants have been fully characterized and prioritizing which *TP53* mutants to model is a challenge given cost and time restraints.”

7. Page 8: the statement that the TP53 mutants tested "do not show any gain-of function activity in the SaOS2 TP53-/- background" appears a generalization not justified by the experimental data presented.

We agree with the reviewers that the studies in the SaOS2 cells which are *TP53* deficient osteosarcoma cells does not agree in vivo studies in our zebrafish embryonal rhabdomyosarcoma model. We would like to note that while the SaOS2 and other *TP53* deficient cells have traditionally used to characterize mutant *TP53* function a significant subset of mutants do not show any phenotypes or show mutant p53 effects only in some and not in other assay systems. Additionally, whether *TP53* mutants can have GOF effects remains controversial and may be tumor type specific. We initially added the SaOS2 data to highlight that the SaOS2 cell based system may not be optimal for the mutants we tested. However, as we have not tested other systems and have not generated knockin alleles for these mutants in ERMS cells lines, we cannot support this assertion and have accordingly removed the SaOS2 data as suggested.

8. The Discussion should reference that the order in which oncogenic mutations occur may impact tumor phenotype and as such, may not be fully reflected in the zebrafish models as presented.

We thank the reviewer for this comment and have added a section to our discussion to address order of the oncogenic mutations and on the limitations of the zebrafish in vivo experimental system we employ. We also include a brief statement of where our studies would fit in addressing an important yet unaddressed need in understanding the function of mutant and wild-type p53 in patients with sarcoma.

Reviewer #3 (Recommendations for the authors):1. In many figures, including Figures 1D, 3C, S1E, and S5G, the plots of Tumor/body size or Tumor area ratio showed very high tumor burden in certain groups, the ratios in some plots were even higher than 0.6. Please make sure your studies complied with the IACUC regulation on the tumor size/weight per animal.

We thank the reviewer for pointing this out. Our studies comply with our IACUC protocol. In the tumor initiation studies in Figure 1B, 1D, 3 B, 3C we are scoring for tumors at the earliest time point 10-20 days at which time in the tp53-/- animals that initiate tumors are removed from the analyses. An important criteria is if the animals are experiencing any distress or are showing differences in their ability to swim. Since the muscle is the major tissue in the zebrafish at 10-20 days most of the tumor burdened animals are in-distinguishable from the no tumor animals under the naked eye and fluorescence is the only way to distinguish the difference. We also present new figure 5 to follow medulloblastomas that occur in the zebrafish brain. In these experiments as the tumors are slower growing and unexpected, we allow the tumors to grow a little bit to help with the histology before we euthanize the animals. In the adult animals (Figure S5G now S4D,H), we again are using fluorescence area and intensity to estimate relative size and growth rates. In these animals the tumors are again on a subset of the normal tissue and externally visible only if they grow locally. In all our experiments death is not an end point and any animal experiencing distress is immediately euthanized.

2. Although the SaOS2 cell line is a commonly used assay system to study effects of TP53 variants, the data generated in this cell line were not consistent at all with those obtained from the in vivo zebrafish studies. Could authors discuss this inconsistency? Can authors use ERM cell line(s) instead of the SaOS2 osteosarcoma cell line to assess the activities of the TP53P153△, TP53C176F and TP53Y220C variants? In addition, engineering the cell lines with these exact mutations, not overexpression, would be helpful to accurately characterize their activities in vitro.

We agree with the reviewers that the studies in the SaOS2 cells which are *TP53* deficient osteosarcoma cells do not agree with in vivo studies in our zebrafish embryonal rhabdomyosarcoma model. We would like to note that while the SaOS2 and other *TP53* deficient cells have traditionally used to characterize mutant *TP53* function a significant subset of mutants do not show any phenotypes or show mutant p53 effects only in some and not in other assay systems. Additionally, whether *TP53* mutants can have GOF effects remains controversial in some tumors for example leukemia (Boettcher et al., 2019). However, as we have not tested other cell systems and have not generated knockin alleles for these mutants in ERMS cells lines, we cannot fully support this assertion and have accordingly removed the SaOS2 data as suggested.

3. Coinjection of either TP53P153△ or TP53Y220C mutant with kRASG12D can increase development of ERMS in the head musculature. However, these two mutants behave very differently in many aspects, such as in kRASG12D-induced tumor initiation (Figures 5B vs. 6C) or in cell survival of ERMs (Figures 5F vs. 6I). Can authors provide clearer explanation on how these two different mutants promote more ERM tumorigenesis in the head musculature? Since it has been hypothesized that both TP53P153△ and TP53Y220C mutants might cause p53 instability, it would be interesting to test whether ZMC1 can stabilize these mutants leading to inhibition of head tumor development. In addition, whether Hedgehog signaling pathway could be specifically activated in the head tumors with overexpression of kRASG12D and TP53P153△ or TP53Y220C?

We thank the reviewer for these comments and have addressed them with the following four sets of experiments.

We tried to generate *TP53^P153^*^Δ^ in the syngeneic CG1; tp53^-/-^ background but unexpectedly discovered that *TP53^P153^*^Δ^ can collaborate with kRASG12D and tp53 loss to initiate medulloblastomas with a sonic hedgehog signature. See Text and new Figure 5 “We generated head ERMS tumors expressing *TP53^P153^*^D^ in the syngeneic CG1; tp53^-/-^ background. We expanded 3 head tumors expressing *TP53^P153^*^Δ^ and also 3 tp53^-/-^ tumors (1 head and 2 trunk tumors), sorted pure secondary tumors and performed RNAseq analyses across both groups. Rather unexpectedly, the head tumors we isolated were all pro-neural brain tumors with gene signatures consistent with the sonic hedgehog group medulloblastoma. To address how frequent the medulloblastoma tumors were, we generated additional tumors with the following genotypes kRAS^G12D^/ tp53^-/-^, kRAS^G12D^/ *TP53^P153^*^Δ^, tp53^-/-^ and kRAS^G12D^/ TP53^C176F^, tp53^-/-^. We identified 5 out 21 head tumors had histology consistent with medulloblastoma in the kRASG12D/ *TP53^P153^*^Δ^ ,tp53-/- group, the rest were all head ERMS tumors (<24% of all head tumors). In contrast only 1 out of 20+ kRAS^G12D^/ tp53^-/-^ tumor had a medulloblastoma tumor and we did not see any head tumors in the kRAS^G12D^/ *TP53^C176F^*/tp53^-/-^ group.In a second set of experiments (See Figure 7A) we asked if similar to the mouse model of hedgehog driven head and neck rhabdomyosarcoma if the head and trunk tumors are molecularly different. RNAseq analyses comparing the CG1; tp53^-/-^ ERMS head tumor with the two trunk tumors and show that the head tumors express genes consistent with head musculature including *tbx1, dlx3b, dlx4b* and *ptch2* and lower expression of other myogenic genes including.In the mouse model of hedgehog driven head and neck RMS, the authors suggest that head and neck tumor arise from Kdr-positive endothelial precursors (Drummond et al., 2018). The cell of origin of zebrafish ERMS tumors is not known but we show that zebrafish ERMS show tumor propagating gene expression signatures consistent with an activated satellite cell (Ignatius et al., 2012). Mutant p53 can activate *KDR* expression.

Genomic analyses of the osteosarcoma from the patient with the germline *TP53^P153^*^*Δ*^ mutation identified that *KDR* was amplified. Further, we confirmed if KDR was expressed via western blot and IHC analyses on PDX that we generated from the patient at autopsy (see Figure 7B, C). Lastly using the zebrafish ERMS model we tested if ablation of *kdr* in zebrafish would significantly reduce or eliminate head ERMS tumors. Our results show that ablating kdr and not control gene *mitfa* resulted in loss of overall tumor initiation and even more significantly head ERMS tumors (Figure 7D-J).

We tried to generate a cell line from the Osteosarcoma PDX expressing *TP53^P153^*^*Δ*^ and have failed thus far. However, we were able to identify that the ewing sarcoma cell line EW8 expressed the *TP53^Y220C^* mutation. We therefore determined if there were differences in the effects on p53 and p21 protein expression of published dosing of ZMC1 and Pikan083 on EW8 cell that has the *TP53^Y220C^* mutation and show that Pikan083 can increase p53 and p21 expression (Author response image 1). We also compared EW8 cells to RMS RD cells that harbors the *TP53^R248W^* mutation and with RMS SMS-CTR cells as controls as they are *TP53* deficient. Our results show that ZMC1 can stabilize p53^R248W^ but to a much lesser extent p53^Y220C^, while Pikan083 stabilizes p53^Y220C^ but not p53 ^R248W^. We also assessed if stabilization of p53 resulted in induction of p21 expression. And while our data holds true for RD and EW8 cells, we unexpectedly find that p21 is induced in SMC-CTR cells treated with both ZMC1 or Pikan083 indicating that p21 can also be induced via p53 independent mechanisms in RMS cells.We also added the data from a recent publication that showed head and neck RMS tumors have a higher proportion of *TP53* mutations compared to other tumor sites (Shern et al., 2021).

(Author response images 2, 3). This data is consistent with our data in our zebrafish ERMS model where p21 is induced *tp53^-/-^* ERMS compared to tumors expressing wt *tp53*. These data also indicate that ZMC1 and Pikan083 can have different effects on p53^Y220C^ and p53^R248W^ mutant protein expression.

**Author response image 1. sa2fig1:** Addition of Pikan083 to EW8 (p53^Y220C^) ewing sarcoma cells results in increased p53 and p21 expression. p53 and p21 expression in EW8 cells treated DMSO or with 1 and 5 μM ZMC1 or 50 and 100 μM Pikan083. Β-tubulin was used as a loading control.

**Author response image 2. sa2fig2:** Addition of ZMC1 and Pikan083 has differential effects on p53 and p21 expression in EW8, RD and SMC-CTR cells. EW8 (p53^Y220C^), RD (p53^R248W^) and SMS-CTR (p53 null) cells treated with 5 µM ZMC1 (top panel) or 100 µM Pikan083 (bottom panel). GAPDH was used as a loading control.

**Author response image 3. sa2fig3:** p21 expression across commonly used rhabdomyosarcoma cell lines. Rh18, Rh36 are WT for p53. RD, JR1, Rh28, Rh30, Rh41 are p53 mutant and SMS-CTR are p53 deficient (Hinson et al., 2013). Β-tubulin was used as a loading control.

e) We also added the data from a recent publication that showed head and neck RMS tumors have a higher proportion of *TP53* mutations compared to other tumor sites (Shern et al., 2021).

4. TP53C176F coinjection with kRASG12D in p53-/- fish can reduce kRASG12D-induced tumorigenesis (Figure 5B) and increase apoptosis (Figure 5F), suggesting TP53C176F functions as a hypomorphic mutant. Can authors examine whether the p21 expression is upregulated in the ERMs with TP53C176F overexpression, which would provide further mechanistic insight on the hypomorphic effect of the TP53C176F mutant.

As requested qPCR was performed on newly generated primary tumors. “To assess the effects of p53 variant expression on transcription, we performed qPCR analyses comparing *tp53^+/+^, tp53^-/-^* and *tp53^-/-^* + *TP53^C176F^* and *tp53^-/-^* + *TP53^P153^*^*Δ*^ ERMS and p53 target genes including *baxa, cdkn1a*, *gadd45a*, *noxa* and *puma/bbc3*. We found that both *tp53^-/-^*+*TP53^C176F^* and *tp53^-/-^* + *TP53^P153^*^*Δ*^ tumors displayed increased *bbc3* expression compared to *tp53^-/-^* or *tp53^+/+^* tumors, but had no difference with respect to *cdkn1a*, *gadd45a* and *noxa* expression. We also found that *tp53^-/-^* + *TP53^C176F^* and *tp53^-/-^* + *TP53^P153^*^*Δ*^ tumors expressed higher levels of *cdkn1a* compared to *tp53^+/+^* ERMS; however, rather unexpectedly *tp53^-/-^* tumors also express higher levels of *cdkn1a* compared to *tp53^+/+^* controls (Figure 4 Supplement 3).”

[Editors' note: further revisions were suggested prior to acceptance, as described below.]

There is strong consensus that the manuscript has been substantially improved but there are some remaining issues that need to be addressed, as outlined below:First, there appears to be a missing item – Supplemental Table I.

We thank the reviewer for pointing this out and apologize for this oversight on our part. While the two versions of the manuscript we submitted have Supplemental Table1. The version generated by the *eLife* server and that was sent out for review did not have Supplemental Table 1.

All tumors were generated by co-expressing Human *TP53* wild-type or mutant alleles along with kRAS^G12D^ and a fluorescent reporter in the *tp53^-/-^* zebrafish background.

Second, the new findings about the occurrence of medulloblastoma in the tp53P153 model should be mentioned in the Abstract.

We have edited our abstract to include changes requested by the reviewers/editors and also added a line stating that *TP53^P153Δ^ when expressed in the kRAS^G12D^*-driven ERMS-model can also initiate medulloblastoma tumors.

*“TP53^P153^*^*Δ*^ unexpectedly also predisposes to hedgehog expressing medulloblastomas in the *kRAS^G12D^*-driven ERMS-model.”

Reviewer #1 (Recommendations for the authors):Overall, the revised manuscript addressed the essential revisions and the authors have responded to the reviewer's requests. The revised manuscript also presents two new analyses, which weren't mentioned in the abstract or introduction. A new Figure 5 and supporting figures show identification of medulloblastoma in addition to head ERMS arising in the tp53P153delta ERMS model. A new Figure 7 shows experiments to test the dependence of the tp53P153delta ERMS tumor induction on Kdr, which is overexpressed in patient osteosarcoma and the zebrafish model. The Kdr gene and function of the protein could be better introduced, and the experiment have stronger experimental validation as described below. But the kdr experiment illustrates how this model can be used to investigate the in vivo mechanisms by which patient tp53 variants drive ERMS tumorigenesis.Some issues were not adequately addressed: the statement that the model is high throughput has not been removed from the discussion.

We thank the reviewer for pointing that oversight and have removed all references to the zebrafish model being a high-throughput platform. We instead highlight the fact that we characterize zebrafish wild-type *tp53* function and also the human *TP53* wild-type allele and three patient specific alleles using the zebrafish ERMS in vivo model.

“Third, we utilized our robust zebrafish ERMS model to assign function to three *TP53* mutants whose effects in ERMS were previously unknown.”

Figure 2 A, B quantification data was not included and it's still unclear what the numbers in those panels reflect,

Figure 2 A. For this analysis we counted Edu+ cells from an entire tumor section (dsRED or GFP+ cells) and normalized Edu+ counts to the total number of DAPI positive cells. Each section represents one ERMS tumor so for this analysis we assessed tumors from multiple tumor burdened zebrafish and these counts were used for statistical analyses. Edu and DAPI counts are available in Supplementary excel files of all the raw data which we have uploaded on the Dryad site. In Figure 2A we show one representative tumor each of Edu+ labeled cells in tumors generated in *tp53^+/+^* or *tp53^-/-^* (null) zebrafish. We also added a third panel with a plot showing % of Edu-positive cells in the different tumor sections. We have also updated our figure legend and methods section to clearly state how the analysis was performed.

Figure 2 B. Our earlier analyses grouped counts from multiple sections. We agree that these counts can be confusing to interpret the data in the images provided. Accordingly, we have updated our counts to reflect the counts in the image shown. For each tumor, we imaged a single confocal section at 200x magnification. We then counted pHH3-positive cells in each confocal section. Counts (sections) from multiple tumor burdened animals per genotype were then used to calculate absolute numbers and for statistical analyses of cells per fixed area undergoing mitosis. We also added a third panel with a plot showing the total number of pHH3-positive cells in the different tumor sections. The counts are available in Supplementary excel files 2A and 2B which we have uploaded on Dryad site.

and new RT-qPCR on p53 target gene expression contained contradictory or inconsistent results. Different tp53 variants did not lead to consistent elevation of apoptotic genes bax, bbc3, noxa, – tp53C176F and tp53P153delta induced expression of bbc3, but not noxa or bax; tp53Y220C increased noxa and cell growth arrest gene gadd54a; cell cycle arrest gene cdkn1a was elevated in all variant models including the tp53 null. A discussion of this data, in the context of demonstrating that the tp53 variants retain some level of transcriptional activity, was not included.

We thank the reviewer for this feedback. We have included a new section in our discussion to address differences in p53 target genes expressed in the tumors with the different *TP53* mutants in the tp53^-/-^ background.

The experiment to demonstrate ERMS tp53P153delta is dependent on elevated kdr expression via somatic CRISPR RNP kdr targeting did not contain much detail. A more thorough analysis of kdr deletion/mutagenesis in the tumors.

We thank the reviewer for this feedback. We have added more details to introduce this experiment including the design of the targeting guide RNAs and why we picked *mitfa* as an experimental control. We also show that we assessed cutting of DNA in 5 of the 12 *kdr* group tumors obtained using DNA heteroduplex assays to show that we get the expected large deletion and also multiple other indels and an almost complete loss of the wild-type band (see Figure 7 Supplement 1). Finally, we cloned the PCR products from *kdr* ablated tumors and sequenced several clones and our analyses shows that our guide RNAs are highly active.

Figure 7 Supplement 1 does not contain enough detail explaining how targeted mutagenesis was induced, documented and measured.

We have added the following to the Methods section and also better annotated Figure 7 Supplement 1.

“Strategy and quantification of mosaic ablation of *kdr* and *mitfa* in zebrafish ERMS tumors using CRISPR/Cas9 reagents.

Using Benchling, three high scoring CRISPR guides (with minimal off-target scores) were selected to target DNA sequences in exons 12 and 13 of *kdr* that encode parts of two Ig repeats in the extracellular domain (aa 535 to 647; InterPro domain IPR003599). Efficient nuclease activity of guides 1 and 3 would result in a 334 bp deletion leading to a predicted frameshift and premature termination of the polypeptide sequence. When this larger deletion does not result from synchronous CRIPSR nuclease activity of guides 1 and 3, similar consequences are likely to result due to independent indels generated from nuclease activity and NHEJ repair at each of the three individual target sites of guides 1, 2 and 3. Since this region is upstream of the transmembrane domain (aa 773-795), we predict that indels or deletions leading to frameshift mutations would, at best, produce a partial extracellular domain containing polypeptide incapable of transducing signals into the cell.

Using PCR and TBE-PAGE, the presence of deletions were visualized by small PCR amplicons corresponding to a loss of 334 bp in a 641 bp wildtype PCR product (deletion mutant amplicon = 212 bp). In addition, presence of heteroduplex products were observed via PAGE indicating the presence of indels. Indels were confirmed by cloning PCR products and sequencing individual clones. Shown in Figure 7 Supplement 1D are indels +21bp, -7 bp at the g1 site, 7 bp deletion at the g2 site and 25 bp deletion at the g3 site. The effectiveness of the three sgRNA/CRISPR guides by way of the spectrum of mutations they cause suggest that despite the mosaicism of the mutations in sgRNA-Cas9 injected embryos, tumors that arise harbor a relatively high frequency loss of function mutations in the *kdr* gene, in all tumors. *mitfa* was selected as a control gene because loss-of-function mutations can be easily screened by looking for the loss of pigmentation in developing larvae using brightfield microscopy. In addition to not having an effect on embryo survival and development, it obviates the need to obtain sequence data to confirm the presence of loss of function mutations in the embryos.”

kdr transcript knockdown measured by RT-qPCR, would validate the conclusion that Kdr drives oncogenesis in the ERMS tp53P153delta model.

Given that *kdr* functions cell autonomously as suggested by (Drummond et al., 2018), where upon trans-differentiation of endothelial cells to rhabdomyosarcoma tumors there is a loss of *kdr* expression. Accordingly, analyses of *kdr* mRNA expression in sorted tumors would be required to address cell autonomous requirements. This is technically challenging due difficulties sorting small ERMS tumors and limited number of head tumors obtained in the *kdr* mutant group. Furthermore, the expected result would be that there is no difference in *kdr* expression in control or *kdr* mutant/loss of function tumors. Our experiments are not designed to address whether the effect we observe is cell autonomous or in the supporting vasculature i.e. effects on tumor angiogenesis. While this is an important question, we believe that this question is beyond the scope of the current study. We incorporate a statement in the manuscript stating the limitation of our analyses.

Reviewer #2 (Recommendations for the authors):The authors have done an admirable job responding to previous reviews, resulting in a manuscript that is more logically constructed and better focused in the data presented.A few residual comments:1. I was not able to find the Supplemental Table that was suggested by reviewers and referred to in the Response to Reviewers that summarizes and highlights the main findings and differences between wild type and various tp53 zebrafish mutants described in the manuscript.

We thank the reviewer for pointing this out and apologize for this oversight on our part.

All tumors were generated by co-expressing Human *TP53* wild-type or mutant alleles along with kRAS^G12D^ and a fluorescent reporter in the *tp53^-/-^* zebrafish background.

2. The choice of analyzing the TP53 P153 mutation remains a bit unusual given its rarity and identification as a germline mutation in an aggressive osteosarcoma, given the focus of this manuscript on ERMS (especially now with removal of the osteosarcoma cell line data which contrasted with the zebrafish data).3. Moreover this P153 mutation leads to Shh medulloblastoma tumors in a subset of zebrafish. This is new data that is interesting, but a bit of a distraction from the focus of the manuscript on ERMS.

We thank the reviewer for this comment but would like to state our reasoning for including the *TP53^P153^*^*Δ*^ mutant analyses. This whole project was initiated because of a patient in our clinic with an aggressive osteosarcoma who responded very poorly to therapy. The patient had a rare germline *TP53* variant of unknown significance (Dr. Amanda Lipsitt attending physician-Second author). The rare *TP53* variant and the patient history where the mom also succumbed to osteosarcoma suggested to us that this may be a gain of function mutant. Assessing the literature, we were surprised that for such a well-known gene, the majority of mutations in sarcoma (osteosarcoma and rhabdomyosarcoma) were uncharacterized and had a different spectrum from the mutations in adult cancers and from those modelled in mice. Since a robust osteosarcoma model in zebrafish is lacking, we decided to test if the *TP53* variant was able to show different effects from the *tp53* null background in our zebrafish ERMS model. The advantage of this model is that tumor initiation occurs as early as 7-10 days and in most analyses majority of the tumors initiate by 60 days. Finally, we unexpectedly found that >97% of animals initiate tumors in the tp53 null background. This enabled us to assess differences in tumor initiation and the retention of wild-type function in *TP53* variants. However, to understand if our system worked, we had to define human wild-type *TP53* and at least one other mutant that was present in a patient with ERMS. These were our controls.

Our unbiased approach has revealed that the zebrafish ERMS model can be used to assess effects on tumor initiation, apoptosis, proliferation, site of tumor initiation and also in the rag2-kRAS^G12D^ ERMS model the initiation of medulloblastoma tumors. We also find that the different *TP53* mutants/variants can differentially regulate the expression of well-known p53 direct transcriptionally regulated genes *puma, noxa, baxa* and *gadd45a*. Specifically, we find that the *TP53^153^*^*Δ*^ is a pathogenic gain of function mutant that predisposes to head ERMS and also the hedgehog-positive sub-type of medulloblastoma tumors. *TP53^153^*^*Δ*^ also directly or indirectly regulates *kdr* expression to promote head ERMS formation. Medulloblastoma initiation in the rag2-kRAS^G12D^ ERMS model has never been reported and strongly supports our finding that *TP53^153^*^*Δ*^ displays gain-of-function effects with respect to site of ERMS initiation and the initiation of medulloblastoma tumors.